# AN EMPIRICAL INVESTIGATION OF THE ROLE OF PRE-TRAINING IN LIFELONG LEARNING

## ABSTRACT

The lifelong learning paradigm in machine learning is an attractive alternative to the more prominent isolated learning scheme not only due to its resemblance to biological learning, but also its potential to reduce energy waste by obviating excessive model re-training. A key challenge to this paradigm is the phenomenon of catastrophic forgetting. With the increasing popularity and success of pre-trained models in machine learning, we pose the question: What role does pre-training play in lifelong learning, specifically with respect to catastrophic forgetting? We investigate existing methods in the context of large, pre-trained models and evaluate their performance on a variety of text and image classification tasks, including a large-scale study using a novel dataset of 15 diverse NLP tasks. Across all settings, we observe that generic pre-training implicitly alleviates the effects of catastrophic forgetting when learning multiple tasks sequentially compared to randomly initialized models. We then further investigate *why* pre-training alleviates forgetting in this setting. We study this phenomenon by analyzing the loss landscape, finding that pre-trained weights appear to ease forgetting by leading to wider minima. Based on this insight, we propose jointly optimizing for current task loss and loss basin sharpness in order to explicitly encourage wider basins during sequential fine-tuning. We show that this optimization approach leads to performance comparable to the state-of-the-art in task-sequential continual learning across multiple settings, without retaining a memory that scales in size with the number of tasks. [1]

## 1 INTRODUCTION

The contemporary machine learning paradigm concentrates on isolated learning (Chen & Liu, 2018) i.e., learning a model from scratch for every new task. In contrast, the lifelong learning (LL) paradigm (Thrun, 1996) defines a biologically-inspired learning approach where models learn tasks in sequence, ideally preserving past knowledge and leveraging it to efficiently learn new tasks. LL has the added benefit of avoiding periodical re-training of models from scratch to learn novel tasks or adapt to new data, with the potential to reduce both computational and energy requirements (Hazelwood et al., 2018; Strubell et al., 2019; Schwartz et al., 2020). In the context of modern machine learning where state-of-the-art models are powered by deep neural networks, *catastrophic forgetting* has been identified as a key challenge to implementing successful LL systems (McCloskey & Cohen, 1989; French, 1999). Catastrophic forgetting happens when the model forgets knowledge learned in previous tasks as information relevant to the current task is incorporated. Mitigating or preventing this phenomenon is critical to achieving true LL.

At the same time, transfer learning (TL) has shown impressive results in both computer vision (CV; Zhuang et al. 2021) and natural language processing (NLP) applications (Howard & Ruder, 2018; Peters et al., 2018; Devlin et al., 2019).[2] The modern TL paradigm involves *pre-training* a fixed architecture, like ResNet (He et al., 2016) or BERT (Devlin et al., 2019), using copious amounts of data, and then *fine-tuning* the learnt parameters on target tasks. Given the tremendous success of pre-trained models, there has been increased interest in understanding their role in improving generalization (Erhan et al., 2010; Neyshabur et al., 2020), speed of convergence (Hao et al., 2019),

---

[1] The code is available in the supplemental material.

[2] One of the original motivations for transfer learning was as a way to enable lifelong learning, discussed in a NIPS-95 workshop on "Learning to Learn" (Pan & Yang, 2009).

successful transfer (He et al., 2019; Pruksachatkun et al., 2020), and out-of-distribution robustness (Hendrycks et al., 2020; Tu et al., 2020). Despite these efforts, the role of pre-trained initializations in lifelong learning settings has been under-explored. In contemporary work, it has been shown that pre-trained models can be used as **feature extractors** (i.e., pre-trained weights are frozen) for task-sequential learning (Hu et al., 2021). Because the pre-trained weights are explicitly frozen in this setting, the model undergoes no catastrophic forgetting. In contrast, **fine-tuning** pre-trained weights updates the pre-trained model parameters and is susceptible to severe forgetting. This is typically the most accurate and thus common TL paradigm (Peters et al., 2019), and the one we consider in this work. To the best of our knowledge, there is no prior work systematically analyzing the role of pre-trained initialization on catastrophic forgetting in lifelong learning scenarios.

Figure 1 shows that simply changing the network initialization to generic pre-trained weights can significantly reduce forgetting on the first task when doing sequential training on five tasks. This observation motivates us to ask: **Does pre-training implicitly alleviate catastrophic forgetting, and if so, why?** To answer this question we conduct a systematic study on existing CV and NLP benchmarks and observe that pre-training indeed leads to less forgetting. We also investigate the effect of the type of pre-trained initialization by analyzing the extent to which three pre-trained Transformer language model variants (Sanh et al., 2019; Devlin et al., 2019; Liu et al., 2019) undergo forgetting, observing that increasing the capacity of the model and diversity of the pre-training corpus play an important role in alleviating forgetting. To further stress-test these models under realistic scenarios, we introduce a dataset with 15 diverse NLP tasks and observe a considerable increase in forgetting on this dataset.

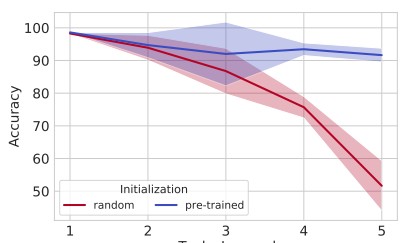

Figure 1: Pre-trained and randomly initialized DistilBERT on Split YahooQA dataset. Performance of the first task visualized over sequential learning of tasks (averaged over 5 runs). Both models start with approximately equal average task accuracy, but pre-trained initialization leads to significantly less forgetting.

We hypothesize that pre-trained weights already have a good inductive bias to implicitly alleviate forgetting. To explain this behavior, we build upon two separate observations — Hao et al. (2019); Neyshabur et al. (2020) show that in the context of TL, pre-trained weights lead to a flat loss basin when fine-tuning on a single task. Mirzadeh et al. (2020b) argues that the geometric properties of the local minima for each task play a vital role in forgetting, and they suggest modifying the hyper-parameters (learning rate decay, batch size, dropout) to promote flat minima.

To verify the above hypothesis, we analyze the loss landscape of the first task as the model is trained sequentially on subsequent tasks. For pre-trained initializations, we see that minima obtained after training on a sequence of tasks still remain in the relatively low loss basin of the first task when compared with random initialization.[3] These observations hint at the flatness of the minima reached in the case of pre-trained initialized models. To quantify the flatness of the loss landscape, we evaluate a sharpness metric (Keskar et al., 2017) and verify that pre-trained weights indeed lead to flat basins in comparison to random weights while training sequentially. These analyses help us showcase that continual training from pre-trained weights induces wide task minima, therefore, implicitly alleviating forgetting. To further mitigate forgetting, we explicitly optimize for flat loss basins by minimizing the current task loss and the sharpness metric. Concretely, we use the Sharpness-Aware Minimization (SAM) procedure (Foret et al., 2021) to seek parameters that lie in the neighborhoods having uniformly low loss values (Section 5) and report improved results across many experimental settings. Our main contributions can be summarized as follows:

- We observe that initializing models with pre-trained weights results in less forgetting compared to random weights despite achieving higher performance on each task. We bolster this observation with a systematic study validating that this behavior persists across applications (NLP, CV) and prominent approaches: Elastic weight consolidation (Kirkpatrick et al., 2017), A-GEM (Chaudhry et al., 2018), and episodic replay (Chaudhry et al., 2019).

---

[3]Linearly interpolating loss across sequentially trained task minima confirms that models initialized with pre-trained weights undergo a more gradual change in loss compared to random initialization (Appendix E.1).

- To understand the role of varying pre-trained initializations, we analyze a suite of pre-trained Transformer language models and showcase that model capacity and diversity of the pre-training corpus do play a role in alleviating forgetting. We also show that sequential training on diverse tasks is still challenging for pre-trained initialized models by introducing a new, more challenging benchmark for LL in NLP consisting of 15 diverse NLP tasks.

- We hypothesize and verify empirically that pre-trained models alleviate forgetting as they have an implicit bias towards wider task minima. The effect of these wider minima is that changes in weights from learning subsequent tasks result in a smaller change to the current task loss, which helps reduce forgetting. We further show that explicitly seeking flat basins during sequential fine-tuning results in even less forgetting.

## 2 PRELIMINARIES

### 2.1 PROBLEM SETUP: LIFELONG LEARNING (LL)

We consider a setup where we receive a continuum of data from different tasks in sequential manner: $(x_1, y_1, t_1), \cdots, (x_i, y_i, t_i), \cdots$. Each triplet $(x_i, y_i, t_i)$ consists of a task descriptor $t_i \in \mathcal{T}$, input data $x_i \in \mathcal{D}_{t_i}$, target labels $y_i \in \mathcal{Y}_{t_i}$ and satisfies $(x_i, y_i) \overset{iid}{\sim} \mathcal{P}_{t_i}(X, Y)$. Following (Chaudhry et al., 2019), we consider an explicit task descriptor $t_i$ because the same input $x_i$ can appear in multiple different tasks but with different labels. For example, given a product review, we could annotate it with sentiment polarity and grammatical acceptability judgments. Given the observed data, our goal is to learn a predictor $f : \mathcal{X} \times \mathcal{T} \to \mathcal{Y}$ where we want to evaluate test pairs $(x, t)$ from previously observed tasks (backward transfer) and the current task at any time during LL of our model.

### 2.2 DATASETS AND TASK SEQUENCES

We perform extensive experiments on widely adopted task-incremental learning benchmarks (Chaudhry et al., 2019; Ebrahimi et al., 2020; Wang et al., 2020) across both CV and NLP domains. These benchmarks help us evaluate our method to be consistent with the literature. Most of the existing works consider Split MNIST, Split CIFAR-10, and Split CIFAR-100 benchmarks, which are homogenous; different tasks in these benchmarks share the same underlying domain. Given the generic nature of the pre-trained initialization, we want to investigate forgetting when subjected to a sequence of diverse tasks. Therefore, we also consider datasets spanning diverse CV and NLP tasks.

**CV benchmarks.** We perform our experiments on 5-dataset (diverse) and Split CIFAR-50 / Split CIFAR-100 (homogenous). **5-dataset** consists of 5 diverse 10-way image classification tasks: CIFAR-10 (Krizhevsky & Hinton, 2009), MNIST (LeCun, 1998), Fashion-MNIST (Xiao et al., 2017), SVHN (Netzer et al., 2011), and notMNIST (Bulatov, 2011). It is one of the largest datasets used for LL experiments (Ebrahimi et al., 2020) with overall $180.9k$ train examples (see Table 4 in the Appendix B for task-specific statistics). **Split CIFAR-50** takes the first 50 classes of the CIFAR-100 image classification dataset (Krizhevsky & Hinton, 2009) and randomly splits them into 5 homogenous 10-way classification tasks. Each task contains $5k/1k$ (train/test) examples. We built this dataset as a homogenous counterpart to 5-dataset by mimicking its task structure (10 classes/task) and the number of tasks. Further, we note that Split CIFAR-50 (10 classes/ task) is more challenging than Split MNIST/ CIFAR-10 (2 classes/ task) because of the more number of classes per task. Therefore, in this work, we prefer Split CIFAR-50 over MNIST/CIFAR-10 for our experimentation. **Split CIFAR-100** splits the CIFAR-100 dataset into 20 disjoint 5-way classification tasks, with each task containing $2.5k/0.5k$ (train/test) examples. Due to a large number of tasks in this dataset, it is regarded as one of the most challenging and realistic CV benchmarks for LL (Chaudhry et al., 2018).

**NLP benchmarks.** We perform our experiments on Split YahooQA (homogenous) and 5-dataset-NLP/ 15-dataset-NLP (diverse). **Split YahooQA** consists of 5 homogenous 2-way classification tasks and is built from a 10-way topic classification dataset (Zhang et al., 2015) by randomly splitting topics into different tasks. Each task includes around $279k/12k$ train/test examples. **5-dataset-NLP** consists of text classification datasets (Zhang et al., 2015) from 5 diverse domains: AGNews, Yelp, Amazon, DBPedia, and YahooQA. Following the data processing procedure mentioned in (de Masson d'Autume et al., 2019), we have $115k/7.6k$ (train/test) examples per task. **15-dataset-NLP**: In order to study the role of different pre-trained initializations in LL, we introduce 15-dataset-

NLP, a novel suite of diverse tasks for LL. It consists of 15 text classification tasks covering a broad range of domains. Although there exists 5-dataset-NLP benchmark, we show that our introduced benchmark proves more challenging than the previous setup (see Section 3.3). 15-dataset-NLP consists of single sentence or sentence pair classification tasks: CoLA (Warstadt et al., 2019), BoolQ (Clark et al., 2019), SST-2 (Socher et al., 2013), QQP[4], YahooQA (Zhang et al., 2015), Yelp (Zhang et al., 2015), Event Factuality (Poliak et al., 2018), Argument Aspect Mining (Stab et al., 2018), Explicit Discourse Marker Prediction (Prasad et al., 2019; Kim et al., 2020), QNLI (Wang et al., 2018), Rocstory (Mostafazadeh et al., 2016), MNLI (Williams et al., 2018), SciTail (Khot et al., 2018), Implicit Discourse Relation Classification (Prasad et al., 2019; Kim et al., 2020), and Emotion Detection (Saravia et al., 2018). For more details on this dataset, see Appendix B.

**Task sequences.** One of the desiderata of a LL method is to be robust to different task sequences as task ordering is unknown beforehand. Hence, we run all of our experiments with 5 random task sequences and report average performance (see Appendix B.1 for task sequences).

## 2.3 EVALUATION

Let $s_{t,\tau}$ denote the accuracy on the task $\tau$ after training on task $t$. After model finishes training on the task $t$, we compute the **average accuracy** ($A_t$), **forgetting** ($F_t$) and **learning accuracy** ($LA_t$) metrics as proposed by Lopez-Paz & Ranzato (2017); Riemer et al. (2019). $F_t$ (also referred to as backward transfer) measures the influence of learning task $t$ on the performance of all previously seen tasks $\tau, (1 \leq \tau < t)$. As the model learns multiple tasks in the sequence, we hope that knowledge acquired during LL should aid the learning of new tasks (forward transfer). $LA_t$ measures the learning capability when model sees the new task $t$ (indirectly measuring forward transfer). Say we learn the $t^{\text{th}}$ task, then $A_t$, $F_t$ and $LA_t$ are defined as follows:

$$A_t = \frac{1}{t}\sum_{\tau=1}^{t} S_{t,\tau} \qquad F_t = \frac{1}{t-1}\sum_{\tau=1}^{t-1}\max_{\tau' \in \{1,\cdots,t-1\}}(S_{\tau',\tau} - S_{t,\tau}) \qquad LA_t = \frac{1}{t}\sum_{\tau=1}^{t} S_{\tau,\tau} \qquad (1)$$

## 2.4 METHODS

We compare our method with state-of-the-art baselines (Chaudhry et al., 2019; Mirzadeh et al., 2020b). We first consider the **finetune (FT)** approach, where we simply fine-tune the model on each task in sequence with no additional constraints on learning. **Elastic weight consolidation (EWC)** (Kirkpatrick et al., 2017) is a regularization-based approach that tries to mitigate forgetting by limiting learning for parameters important to previously learned tasks, as measured by the Fisher information matrix. **A-GEM** (Chaudhry et al., 2018) and **episodic replay (ER)** (Chaudhry et al., 2019) methods augment the base model with episodic memory module which retains examples from the previously seen tasks. Following (Chaudhry et al., 2019), we retain one example per task per class and randomly select examples for storage. **Stable SGD** (Mirzadeh et al., 2020b) controls training dynamics by varying the learning rate, learning rate decay, dropout and batch size (see Appendix A). (Prabhu et al., 2020; Hussain et al., 2021) show that simple ER method outperforms all of the previous methods under realistic task-incremental learning setting and we compare our method with ER.

## 3 DOES PRE-TRAINING IMPLICITLY ALLEVIATE FORGETTING?

Having defined the formal problem definition, evaluation metrics, and methods for alleviating the forgetting phenomenon, in this section we conduct experiments to tease apart the role of pre-training for LL. We are interested in answering the following questions: (Q1) How much does pre-training help in alleviating the forgetting? (Q2) Do pre-trained weights undergo similar forgetting on diverse and homogeneous tasks? (Q3) How do different pre-trained initializations affect forgetting?

**Experimental design.** To answer these questions convincingly, we conduct experiments on the above discussed CV and NLP datasets. We utilize the DistilBERT_{base} (Sanh et al., 2019) architecture for text classification and the ResNet-18 (He et al., 2016) architecture for image classification. To isolate the effect of pre-training, we consider two variants for each of these architectures: pre-trained models (**DistilBERT-PT**, **ResNet-18-PT**) and randomly initialized models (**DistilBERT-R**, **ResNet-18-R**).

---

[4]https://www.quora.com/share/First-Quora-Dataset-Release-Question-Pairs

Table 1: Comparing performance in terms of accuracy, forgetting, and learning accuracy across methods after training on the last task (averaged over 5 runs). ↑ indicates higher, ↓ lower is better. We observe that pre-trained models undergoes significantly less forgetting compared to the randomly initialized models. Augmenting the Finetune baseline with SAM (Section 5) results in performance competitive with SOTA, and augmenting the ER baseline with SAM often outperforms SOTA.

| | w/o Pretraining (ResNet-18-R/ DistilBERT-R) | | | w/ Pretraining (ResNet-18-PT/ DistilBERT-PT) | | |
|---|---|---|---|---|---|---|
| | Accuracy(%) ↑ | Forgetting(%) ↓ | LA(%) ↑ | Accuracy(%) ↑ | Forgetting(%) ↓ | LA(%) ↑ |
| **Split-YahooQA** | | | | | | |
| Finetune | 73.10 (±4.69) | 26.39 (±5.88) | 94.21 (±0.03) | 87.66 (±3.72) | 9.45 (±4.66) | 95.22 (±0.02) |
| Finetune + SAM | 73.45 (±4.04) | 25.94 (±5.04) | 94.20 (±0.03) | 88.53 (±2.81) | 8.37 (±3.52) | 95.22 (±0.00) |
| EWC | 76.06 (±3.09) | 22.69 (±3.88) | 94.21 (±0.02) | 89.52 (±3.35) | 7.13 (±4.20) | 95.22 (±0.02) |
| ER | 77.19 (±3.33) | 21.28 (±4.17) | 94.22 (±0.01) | **89.35 (±0.69)** | **7.33 (±0.90)** | 95.22 (±0.00) |
| ER + SAM | **77.48 (±1.40)** | **20.91 (±1.76)** | 94.21 (±0.02) | 88.98 (±0.69) | 7.78 (±0.91) | 95.20 (±0.04) |
| **5-dataset-NLP** | | | | | | |
| Finetune | 44.27 (±4.97) | 36.66 (±6.27) | 73.59 (±0.07) | 64.28 (±4.53) | 16.73 (±5.67) | 77.67 (±0.06) |
| Finetune + SAM | 45.95 (±4.99) | 34.30 (±6.25) | 73.39 (±0.05) | 66.41 (±2.76) | 13.93 (±3.48) | 77.56 (±0.06) |
| EWC | 48.71 (±4.92) | 31.11 (±6.19) | 73.60 (±0.04) | 66.77 (±3.30) | 13.60 (±4.15) | 77.64 (±0.07) |
| ER | **56.34 (±3.05)** | **21.58 (±3.90)** | 73.60 (±0.10) | 70.17 (±1.60) | 9.39 (±2.04) | 77.68 (±0.06) |
| ER + SAM | 56.27 (±3.93) | 21.45 (±5.03) | 73.43 (±0.12) | **71.07 (±1.21)** | **8.07 (±1.51)** | 77.53 (±0.04) |
| **Split CIFAR-50** | | | | | | |
| Finetune | 42.76 (±3.14) | 23.68 (±1.09) | 66.44 (±2.14) | 86.29 (±1.16) | 7.11 (±0.92) | 93.40 (±0.46) |
| Finetune + SAM | **50.34 (±2.19)** | **14.95 (±2.14)** | 65.30 (±1.20) | **90.45 (±1.11)** | **4.24 (±0.98)** | 94.69 (±0.40) |
| EWC | 45.28 (±2.53) | 20.65 (±1.47) | 65.93 (±1.28) | 86.15 (±0.85) | 7.36 (±0.92) | 93.52 (±0.65) |
| A-GEM | 47.34 (±2.65) | 21.08 (±1.95) | 68.42 (±0.71) | 87.25 (±0.95) | 6.16 (±0.63) | 93.42 (±0.41) |
| ER | 45.76 (±1.76) | 20.63 (±1.41) | 66.38 (±2.65) | 86.16 (±1.05) | 7.14 (±0.84) | 93.30 (±0.46) |
| ER + SAM | 50.75 (±0.53) | 16.88 (±0.85) | 67.63 (±0.68) | 88.44 (±1.26) | 5.96 (±1.14) | 94.40 (±0.26) |
| Stable SGD | 46.02 (±2.33) | 12.07 (±0.42) | 58.09 (±2.45) | 84.06 (±1.89) | 5.16 (±1.61) | 89.21 (±0.73) |
| **5-dataset** | | | | | | |
| Finetune | 33.72 (±2.53) | 51.51 (±2.58) | 85.23 (±1.99) | 57.19 (±5.10) | 38.28 (±5.01) | 95.47 (±0.20) |
| Finetune + SAM | 47.58 (±3.75) | 40.60 (±3.98) | 88.19 (±1.33) | 70.40 (±4.37) | 25.61 (±4.44) | 96.01 (±0.10) |
| EWC | 34.99 (±4.94) | 50.05 (±6.53) | 85.05 (±1.85) | 56.67 (±3.75) | 38.75 (±3.78) | 95.42 (±0.16) |
| A-GEM | 46.05 (±6.75) | 39.48 (±7.13) | 85.17 (±2.53) | 71.98 (±2.27) | 23.00 (±2.27) | 94.98 (±0.22) |
| ER | 50.58 (±4.50) | 35.02 (±5.35) | 85.60 (±1.31) | 70.73 (±1.52) | 24.16 (±1.43) | 94.89 (±0.19) |
| ER + SAM | **60.26 (±3.94)** | **27.32 (±4.05)** | 87.59 (±1.32) | **77.40 (±3.91)** | **18.23 (±3.90)** | 95.63 (±0.18) |
| Stable SGD | 50.20 (±7.04) | 40.31 (±7.81) | 90.51 (±0.97) | 71.32 (±2.71) | 20.53 (±2.49) | 91.86 (±0.84) |
| **Split CIFAR-100** | | | | | | |
| Finetune | 38.89 (±2.20) | 39.11 (±2.02) | 77.96 (±0.96) | 81.98 (±2.97) | 13.83 (±2.55) | 95.81 (±0.50) |
| Finetune + SAM | 48.93 (±4.78) | 28.53 (±4.97) | 77.46 (±0.87) | 88.31 (±1.72) | 8.60 (±1.28) | 96.91 (±0.55) |
| EWC | 37.37 (±1.47) | 40.12 (±1.82) | 77.47 (±1.54) | 81.29 (±2.52) | 14.52 (±2.13) | 95.81 (±0.61) |
| A-GEM | 46.84 (±3.48) | 32.01 (±3.77) | 78.79 (±0.95) | 84.00 (±1.58) | 11.71 (±1.02) | 95.70 (±0.69) |
| ER | 48.60 (±1.86) | 29.84 (±1.32) | 78.10 (±0.66) | 84.41 (±2.17) | 11.36 (±1.71) | 95.75 (±0.53) |
| ER + SAM | **60.53 (±0.45)** | **20.91 (±0.69)** | 81.39 (±0.70) | **88.41 (±0.71)** | **8.61 (±0.20)** | 96.73 (±0.53) |
| Stable SGD | 52.94 (±1.71) | 21.02 (±2.01) | 73.83 (±1.53) | 86.64 (±2.17) | 5.53 (±1.54) | 91.75 (±0.74) |

For our study, we need to ensure that there are as few confounding factors as possible. Therefore, we keep all other hyperparameters the same and vary only the initialization (for more details refer to Appendix A). To measure the severity of forgetting, we ideally want sufficient training samples to ensure either a pre-trained model or randomly initialized model of the same capacity can achieve similar learning accuracy on each task. To control for this behavior we either select a large training corpus whenever available (e.g., 279k examples/task for Split YahooQA) or run our experiments for multiple epochs (5 epochs for CV benchmarks). Finally, the ResNet-18-PT model was pretrained on a subset of Imagenet consisting of 733 classes where classes that were semantically similar to the classes in the CIFAR-100 dataset were removed (more details Appendix B).

## 3.1 HOW MUCH DOES PRE-TRAINING HELP IN ALLEVIATING FORGETTING?

From Table 1, we see that pre-trained models (ResNet-18-PT, DistilBERT-PT) undergo significantly less forgetting in comparison to models with random initializations (ResNet-18-R, DistilBERT-R). This trend holds across all three methods. For text classification tasks (Split YahooQA, 5-dataset-NLP), we see that both models have comparable learning accuracy (LA) and significantly less forgetting for DistilBERT-PT. This can be completely attributed to the pre-trained initialization. Now on 5-dataset, ResNet-18-PT (38.28) undergoes less forgetting when compared to ResNet-18-R (51.51). Specifically, despite task accuracy starting at a higher base for ResNet-18-PT, *the absolute forgetting value is still lower compared to ResNet-18-R models*. Additionally, this effect also holds

when considering a sequentially finetuned pre-trained model (with no additional regularization to alleviate forgetting) to a randomly initialized model trained with LL methods. For example, on 5-dataset-NLP, sequentially finetuning DistilBERT-PT undergoes less forgetting (16.73) compared to competitive ER method (21.58) when applied to DistillBERT-R. This raises an interesting research direction — explicitly focusing on learning generic features while training sequentially apart from just focusing on the forgetting aspect of LL. Further, in Appendix C we discuss task-specific results.

### 3.2 Do pre-trained weights undergo similar forgetting on diverse and homogeneous tasks?

From Table 1, we see that ResNet-18-PT does not undergo a significant amount of forgetting when sequentially fine-tuned on Split CIFAR-50, Split CIFAR-100 (homogenous tasks). On Split CIFAR-50, forgetting is around 7 accuracy points. Surprisingly, the competitive ER method also undergoes a similar amount of forgetting, thereby raising a question about the applicability of these datasets when studying forgetting in the context of the pre-trained models. It may be possible to manually cluster tasks based upon semantic closeness, rendering severe interference to make these benchmarks more challenging (Ramasesh et al., 2020). Given the generic nature of the pre-trained initialization, we ask: What happens when we train the model sequentially on diverse tasks? To answer this question, we conduct experiments on 5-dataset and 5-dataset-NLP. From Table 1, **we empirically observe that pre-trained models are susceptible to forgetting when exposed to diverse tasks**. Particularly, DistilBERT-PT undergoes a 16.73 point drop in accuracy when trained on 5-dataset-NLP. Similarly, ResNet-18-PT undergoes a 38.28 point drop in accuracy when trained on 5-dataset.

### 3.3 How do different pre-trained initializations affect forgetting?

To examine the impact of varying pre-trained initialization on forgetting, we evaluate different pre-trained Transfomer models, DistilBERT (Sanh et al., 2019), BERT (Devlin et al., 2019), RoBERTa (Liu et al., 2019), on text classification tasks. From the previous subsection, we observe that pre-trained models are relatively more susceptible to forgetting on LL of diverse tasks. In response, we conduct a thorough investigation on the 5-dataset-NLP. From Table 2, we observe that when keeping the pre-training corpora the same and increasing the capacity of the model — DistilBERT (66M), BERT-base (110M), and BERT-large (336M) — we observe that larger models undergo less forgetting on sequential training of diverse NLP tasks. Further, to explore the impact of the diversity of the pre-training corpora, we compare BERT-base (110M) with RoBERTa-base (125M). We observe that the RoBERTa-base model performs far superior to BERT-base, thus hinting at the necessity of diverse pre-training corpora to implicitly alleviate forgetting. To stress-test these models, we experiment with the 15-dataset-NLP. We observe that by increasing the number of tasks in the sequence, pre-trained models undergo severe forgetting. Surprisingly, the RoBERTa-base model out-performs BERT-Large despite having many fewer parameters. Empirically, we infer that **diversity of pre-training corpora plays a vital role in easing forgetting during LL of diverse tasks.**

## 4 Exploring the Loss Landscape

To better understand how pre-training reduces forgetting, we perform experiments analyzing where models are situated in the loss landscape after training on each task. We denote model parameters after training on task $k$ as $w_k$. If we define forgetting as the increase in loss for a given task during training (instead of decrease in accuracy), Mirzadeh et al. (2020b) show that the forgetting can actually be bounded by:

$$L_1(w_2) - L_1(w_1) \approx \frac{1}{2}\Delta w^\top \nabla^2 L_1(w_1)\Delta w \leq \frac{1}{2}\lambda_1^{max}\|\Delta w\|^2 \tag{2}$$

where $L_1(w)$ represents the loss on Task 1 with parameters $w$, $\Delta w = w_2 - w_1$, and $\lambda_1^{max}$ is the largest eigenvalue of $\nabla^2 L_1(w_1)$. The magnitude of the eigenvalues of $L_1(w)$ can be used to characterize the curvature of the loss function (Keskar et al., 2017), and thus $\lambda_1^{max}$ can be thought of as a proxy for the flatness of the loss function (lower is flatter). From Equation 2, we can see that the flatter the minima, the less forgetting occurs in the model.

We hypothesize that the one explanation of improvements from pre-training shown in the previous section might be because pre-training leads to a more favorable loss landscape. Specifically, pre-

Table 2: Comparing performance in terms of average accuracy, forgetting, and learning accuracy for sequential finetuning after training on the last task. ↑ indicates higher is better, ↓ indicates lower is better. All metrics are averaged across 5 runs. Overall, we observe that models pre-trained on diverse corpora (RoBERTa-base) undergo less forgetting across both 5 and 15 diverse tasks. Augmenting the Finetune baseline with SAM (Section 5) results in performance competitive with SOTA, and augmenting the ER baseline with SAM often outperforms SOTA.

| | 5-dataset-NLP (1 epoch) | | | 15-dataset-NLP (1 epoch) | | |
|---|---|---|---|---|---|---|
| | Accuracy(%) ↑ | Forgetting(%) ↓ | LA(%) ↑ | Accuracy(%) ↑ | Forgetting(%) ↓ | LA(%) ↑ |
| **DistilBERT** | | | | | | |
| Finetune | 64.28 (±4.53) | 16.73 (±5.67) | 77.67 (±0.06) | 46.99 (±3.50) | 18.80 (±4.03) | 64.41 (±1.19) |
| Finetune + SAM | 66.41 (±2.76) | 13.93 (±3.48) | 77.56 (±0.06) | 47.51 (±3.12) | 16.46 (±3.84) | 62.53 (±0.78) |
| ER | 70.17 (±1.60) | 9.39 (±2.04) | 77.68 (±0.06) | 53.22 (±3.39) | 13.05 (±4.07) | 65.32 (±0.59) |
| ER + SAM | **71.07 (±1.21)** | **8.07 (±1.51)** | 77.53 (±0.04) | **53.46 (±2.04)** | **11.00 (±3.05)** | 63.07 (±1.04) |
| **BERT-base** | | | | | | |
| Finetune | 67.55 (±2.76) | 13.57 (±3.37) | 78.40 (±0.07) | 52.86 (±2.82) | 19.20 (±3.01) | 70.76 (±0.27) |
| Finetune + SAM | 70.82 (±2.08) | 9.53 (±2.52) | 78.44 (±0.07) | 55.11 (±2.61) | 16.38 (±3.21) | 70.39 (±0.65) |
| ER | 70.61 (±2.14) | 9.74 (±2.61) | 78.40 (±0.07) | 56.39 (±2.93) | 15.70 (±3.03) | 71.03 (±0.38) |
| ER + SAM | **72.97 (±1.54)** | **6.88 (±1.89)** | 78.47 (±0.07) | **57.82 (±1.85)** | **13.65 (±1.91)** | 70.53 (±0.47) |
| **RoBERTa-base** | | | | | | |
| Finetune | 71.43 (±1.67) | 9.52 (±2.03) | 79.04 (±0.07) | 55.48 (±3.10) | 21.02 (±3.01) | 75.10 (±0.52) |
| Finetune + SAM | 72.56 (±1.22) | 7.83 (±1.53) | 78.83 (±0.02) | 57.82 (±1.72) | 15.40 (±1.88) | 72.08 (±1.36) |
| ER | 73.69 (±0.76) | 6.74 (±0.93) | 79.09 (±0.10) | 60.92 (±1.44) | 15.32 (±1.58) | 75.22 (±0.12) |
| ER + SAM | **74.33 (±0.62)** | **5.65 (±0.76)** | 78.85 (±0.02) | **62.14 (±1.54)** | **12.17 (±2.09)** | 73.30 (±0.79) |
| **BERT-Large** | | | | | | |
| Finetune | 70.98 (±2.02) | 10.22 (±2.53) | 79.15 (±0.03) | 53.82 (±1.21) | 23.42 (±1.43) | 75.68 (±0.41) |
| Finetune + SAM | 73.66 (±1.28) | 6.86 (±1.58) | 79.15 (±0.03) | 58.74 (±3.38) | 17.11 (±4.26) | 74.64 (±2.18) |
| ER | 73.46 (±1.15) | 7.17 (±1.44) | 79.20 (±0.07) | 61.06 (±2.55) | 15.06 (±2.87) | 75.12 (±1.21) |
| ER + SAM | **74.57 (±0.62)** | **5.73 (±0.83)** | 79.15 (±0.09) | **61.74 (±1.25)** | **13.73 (±2.77)** | 74.53 (±1.49) |

training results in wider, flatter minima for each task. The effect of these wider minima is that the change in weights from learning on future tasks results in a smaller change on the actual loss for the current task, which leads to less forgetting. We verify this idea in two parts. First we use loss contours and then interpolate between model checkpoints to show that the flat loss basins lead to smaller changes in loss. Next we compute a sharpness metric to show that pre-training leads to flat loss basins. All models analyzed in this section are trained using the Finetune method.

### 4.1 Loss Contour

In Figure 2, we visualize the contours of the test loss for the first task. On those contours, we plot the locations of the model $(w_1, w_2, w_3)$ after training on each of the first three tasks. Pre-training results in significantly wider optima. In fact, as the model is trained on tasks sequentially, the pre-trained model still remains mostly at the same loss level as compared to randomly initialized models, despite moving approximately the same (or even more) euclidean distance away from the original model. For example, in the plot for the pre-trained model on 5-dataset (Figure 2e), the model after the second task ($w_2$) remains at the same Task 1 loss level as after just training on Task 1 ($w_1$). It is a couple of loss levels higher for task 3 ($w_3$). For the randomly initialized model (Figure 2a), the euclidean distances between the model parameter vectors are approximately the same as for the pre-trained model, but the differences in Task 1 loss levels are significantly higher. We provide more instances of these visualizations along with loss interpolation plots in Appendix E.

### 4.2 Sharpness

As another measure of the wideness of the minima, we calculate a sharpness metric (Keskar et al., 2017) for the model on each task as it goes through training. The metric tries to find the maximum value of the loss in the neighborhood of the minima, and calculates the difference between the maximum and the minimum loss value, scaled by the loss value. The maximization is performed in a subspace of the entire parameter space $\mathbb{R}^n$, specified by a projection matrix $A \in \mathbb{R}^{n \times p}$. For our experiments, we randomly sample our matrix $A$ and set $p = 100$ as in Keskar et al. (2017). The neighborhood of the metric is given by:

$$C_\epsilon = \{z \in \mathbb{R}^p : -\epsilon(|(A^+x)_i| + 1) \leq z_i \leq \epsilon(|(A^+x)_i| + 1) \forall i \in \{1 \ldots p\}\} \quad (3)$$

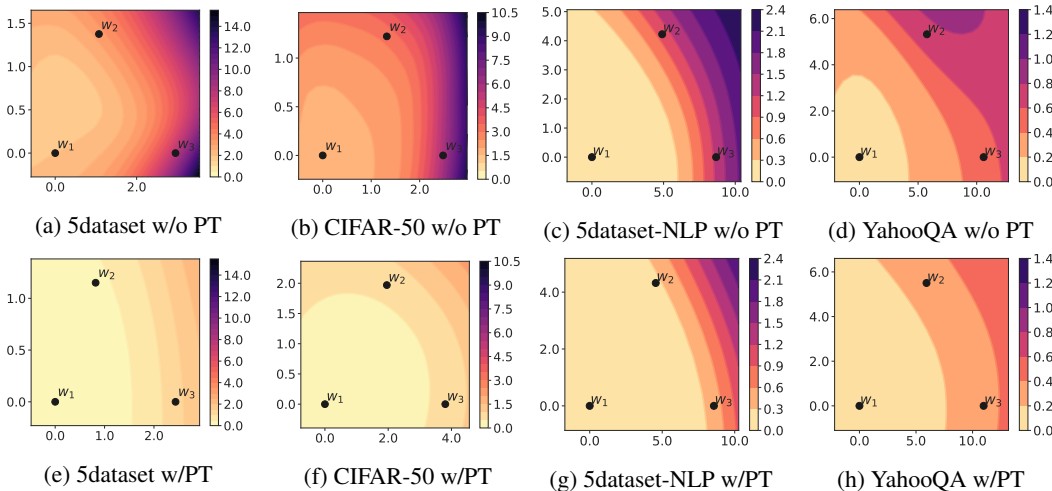

Figure 2: Loss contours for Task 1 on each dataset. Each contour shows the location of the model parameters after training on each of the first three tasks. The top row shows contours for randomly initialized models and the bottom row shows contours for pre-trained initialized models.

Table 3: Average sharpness (lower is flatter) of minima across tasks in a 100 dimensional random subspace. Pre-training (PT) leads to flat minima for each task in training by an order of magnitude.

| Dataset | Method | $\epsilon = 5 \times 10^{-4}$ | | $\epsilon = 10^{-3}$ | |
|---|---|---|---|---|---|
| | | w/o PT | w/ PT | w/o PT | w/ PT |
| 5-dataset | Finetune | 2.07($\pm$0.56) | 0.09($\pm$0.02) | 5.65($\pm$1.64) | 0.20($\pm$0.04) |
| | Finetune + SAM | 0.69($\pm$0.18) | 0.11($\pm$0.01) | 1.82($\pm$0.41) | 0.25($\pm$0.02) |
| Split CIFAR-50 | Finetune | 2.26($\pm$0.68) | 0.18($\pm$0.06) | 6.13($\pm$1.45) | 0.44($\pm$0.11) |
| | Finetune + SAM | 0.73($\pm$0.14) | 0.24($\pm$0.02) | 1.99($\pm$0.32) | 0.55($\pm$0.04) |

where $A^+$ is the pseudo inverse of $A$, $x$ is the parameter vector and $\epsilon$ is a hyperparameter controlling the size of the neighborhood. The metric is then defined as:

$$\phi_{x,f} := \frac{(\max_{y \in C_\epsilon} f(x + Ay)) - f(x)}{1 + f(x)} \times 100 \tag{4}$$

where $f(x)$ denotes the loss value with parameters $x$. We calculate the sharpness metric at $\epsilon = 5 \times 10^{-4}$ and $10^{-3}$. After training on each task, we compute the sharpness values of the minimum reached by the model on that task. We then take the sharpness value of the run to be the mean of the values across the sequence of tasks. We present the mean and standard deviation across 5 runs. The sharpness values for **5-dataset** and **Split CIFAR-50** are presented in Table 3 and for rest of the datasets see Table 6 and Table 7 (in Appendix D). We see that for all datasets, the sharpness values for the pretrained initialized models are significantly lower than the values for the randomly initialized models.

## 5 LIFELONG LEARNING WITH SHARPNESS AWARE MINIMIZATION (SAM)

In the previous section, we looked at how initialization plays an important role in overcoming forgetting as pre-trained initializations favor flat loss basins. On the other hand, (Mirzadeh et al., 2020b) suggests modifying the training regime by varying learning rate decay, batch size, and dropout regularization such that inherent noise in the stochastic gradients leads to flat basins in the loss landscape. However, the procedure for tuning these hyperparameters before continual learning of tasks is ill-defined, thereby rendering their strategy less helpful and quite expensive. Doing a sweep over just the hyperparameter values suggested in the original paper requires 48 separate runs.

Furthermore, these hyperparameters cannot be reused across different architectures and datasets, and thus we need to do even more sweeps. To alleviate these issues, we propose to explicitly seek flat basins while sequential fine-tuning of the pre-trained weights. Towards this objective, we employ the Sharpness-Aware Minimization (SAM) procedure (Foret et al., 2021) that seeks parameters that lie in the neighborhoods having uniformly low loss values by jointly minimizing the task loss value and sharpness metric. SAM defines the sharpness of the loss function $f$ at parameters $x$ as:

$$\max_{||\epsilon||_2 \leq \rho} f(x + \epsilon) - f(x) \tag{5}$$

where maximization region is an $\ell^p$ ball with radius $\rho$ for $p = 2$ in Equation (5). SAM problem can be defined in terms of the following minimax optimization:

$$\min_x \max_{||\epsilon||_2 \leq \rho} f(x + \epsilon) + \lambda ||x||_2^2 \tag{6}$$

The gradient of the result of the maximization problem can be approximated as

$$\nabla_x \max_{||\epsilon||_2 \leq \rho} f(x + \epsilon) \approx \nabla_x f(x)\Big|_{x + \hat{\epsilon}(\mathbf{x})} + \frac{\partial \hat{\epsilon}(\mathbf{x})}{x} \nabla_x f(x)\Big|_{x + \hat{\epsilon}(\mathbf{x})}, \tag{7}$$

where

$$\hat{\epsilon}(\mathbf{x}) = \rho \, \text{sign}(\nabla_x f(x)) \Big( \frac{|\nabla_x f(x)|}{||\nabla_x f(x))||} \Big)^{\frac{1}{2}} \tag{8}$$

To make the optimization simpler, the second order term in the gradient is dropped, leaving us with

$$\nabla_x \max_{||\epsilon||_2 \leq \rho} f(x + \epsilon) \approx \nabla_x f(x)\Big|_{x + \hat{\epsilon}(\mathbf{x})} \tag{9}$$

The full derivation of this gradient can be found in Foret et al. (2021).

From the results in Tables 1 and 2, we can see that SAM results in a consistent improvement in performance over non-SAM counterparts. Simply adding SAM to the sequential finetune baseline makes the approach competitive with, and sometimes outperform, state-of-the-art baselines like ER and Stable SGD, and with minimal hyperparameter tuning required (we used the default value of the $\rho = 0.05$ parameter for all experiments). Finally, since SAM is simply an optimizer, it can be combined with existing continual learning methods. Combining SAM with ER results in a method that outperforms all existing baselines.

## 6 RELATED AND FUTURE WORK

We focus on several closely related lines of work studying the optimization and loss landscapes of deep learning and lifelong learning models here, and discuss some more distant related work in Appendix F. Hao et al. (2019) show that for single-task generalization, pre-training leads to wider optima for BERT models. Keskar et al. (2017) explore how larger batch sizes lead to sharper minima and worse generalization in the single-task learning setting. Mirzadeh et al. (2020b) look at how catastrophic forgetting can be impacted by the training regime, and show that certain hyperparameter settings produce wider minima which lead to less catastrophic forgetting. Finally, Mirzadeh et al. (2020a) compare minima that result from multitask learning and continual learning, and show that the the minima resulting from continual learning are linear mode connected to the optimal sequential multitask minima, but not to each other, which results in forgetting and a corresponding drop in performance. These works all either explore the relation between pre-training and flatness of minima in single-task settings, or between flatness of minima and model generalization capability. We extend this line of work by examining whether benefits from pre-training can persist across training on several tasks, assessing the effects of pre-training on loss landscapes over the course of lifelong learning, and validating a hypothesis explaining the effects of pre-training on lifelong learning.

Based on our findings and results, a potential future work could explore where the multitask minima are in relation to the pre-trained initialization, as Mirzadeh et al. (2020a) show that the sequential multitask minima are linear mode connected to minima after each task in lifelong learning. The flatness of the minima for every model starting from a pre-trained initialization could suggest a way to regularize the sequential training process with the pre-trained initialization such that that the model ends up at the multitask minima. One final takeaway from these results is that lifelong learning methods should focus on creating more general representations instead of simply reducing catastrophic forgetting, as more general representations appear to result in more robust learning.

## 7 ETHICS STATEMENT

Large neural network models are known to mimic and potentially emphasize biases found in the data, including biases that can negatively impact the individuals who interact with these increasingly ubiquitous models. Our work does not analyze this aspect of these models as it pertains to lifelong learning, but without such analysis it is unknown whether lifelong learning could further exacerbate negative societal biases exhibited in these models. We leave this important analysis to future work.

## 8 REPRODUCIBILITY STATEMENT

We include the code, and instructions needed to reproduce the main experimental results in the supplemental material. We specify all the training details including the hyper-parameters for our method and baselines ones. The relevant training details are provided in Sections 3, 4, 5 and Appendix A. Every experiment in the paper was run across 5 random seeds and is reported with error bars (See Tables 1, 2 and Figure 5). We cite all datasets and baselines used as a part of our experiments in Section 2. In the supplemental material, we provide a way to aggregate our new 15-dataset-NLP.

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

## A    IMPLEMENTATION DETAILS

**Vision Experiments**    For all vision experiments, we use the full ResNet-18 He et al. (2016) architecture, with the final linear layer replaced (the number of outputs corresponds to the total number of classes in all given tasks). During inference, only the subset of outputs corresponding to the given task is considered. All images are resized to $224 \times 224$, and normalized with $\mu = (0.485, 0.456, 0.406)$ and $\sigma = (0.229, 0.224, 0.225)$. We used a SGD optimizer with the learning rate set to .01 for all methods (we did a hyperparameter search for both pre-trained and randomly initialized models and found the learning rate $0.01$ resulted in a good learning accuracy for both pre-trained and randomly initialized models). The batch size was set to $10$ for the Split CIFAR-50 and Split CIFAR-100 experiments and $64$ for the 5-dataset experiments. The memory per class for ER was set to $1$, and the $\lambda$ parameter for EWC was also set to $1$.

For Stable SGD, we performed a hyperparameter sweep over the parameters specified in the original paper, namely:

- initial learning rate: [.25 (Split-CIFAR100-Random, Split-CIFAR50-Random, 5data-Random), .1, .01 (Split-CIFAR100-PT, Split-CIFAR50-PT), .001 (5data-PT)]

- learning rate decay: [0.9 (Split-CIFAR50-Random, 5data-Random, Split-CIFAR100-PT), 0.85 (Split-CIFAR100-Random, Split-CIFAR50-PT), 0.8 (5data-PT)]

- batch size: [10 (all), 64]

- dropout: [0.5 (5data-Random), 0.25 (Split-CIFAR100-Random, Split-CIFAR50-Random, Split-CIFAR100-PT, Split-CIFAR50-PT, 5data-PT)]

**NLP Experiments**    For most of the text classification experiments, we use the Transformer architecture based text encoder, DistilBERT-base (Sanh et al., 2019) to encode our input. In a single sentence text classification task, $x_t$ is an input sentence to be classified. In a sentence-pair classification task, concatenation of $x_t^1$ and $x_t^2$ sentences separated by a $[SEP]$ symbol is considered as a input $x_t$. DistilBERT produces a contextual representation of each token in $x_t$ including a special beginning of the sentence token symbol $[CLS]$. We use the representation of the $[CLS]$ symbol from model as features for a linear task classifier. We have a separate classifier for each task. We mainly set hyper-parameters to default implementation from HuggingFace.[5] We use Adam as our optimizer, set dropout $0.1$, the base learning rate to $2e^{-5}$, batch size to $32$ and the maximum total input sequence length after tokenization to $128$. For EWC, we set the regularization strength $\lambda$ to $100$ (as this ended up with comparable LA across other methods) and for ER, following (Chaudhry et al., 2019), the memory per class per task is set to $1$. For SAM, we set $\rho = 0.02$ for all models (random as well as pre-trained) on 5-dataset-NLP and 15-dataset-NLP. For SplitYahooQA we set $\rho = 0.001$.

## B    DATASETS

**ImageNet pre-training corpus**    For a fair comparison between pre-trained and randomly initialized models, we wanted to explicitly control for and remove the overlap between pre-training and downstream tasks. Publicly available ResNet models are pre-trained on ImageNet that overlaps with CIFAR-100 in terms of class labels. Therefore, we make sure that the subset of the ImageNet corpus we use does not have any visually and semantically overlapping classes with the CIFAR-100 dataset. We use the publicly available (Abdelsalam et al., 2021) two-level class hierarchies for ImageNet, where semantically and visually similar labels are grouped under one super-category. We iterate over all CIFAR-100 labels and drop the complete super-category from ImageNet corresponding to each of these labels. For example, CIFAR-100 contains a *castle* class and we have a *building* super-category in ImageNet that contains *castle, palace, monastery, church, etc.*. We remove all building-related labels from our pre-training dataset. In total, we remove 267 classes and pre-train the **ResNet-18-PT** model on the remaining subset of the ImageNet dataset.

---

[5]https://github.com/huggingface/transformers

Table 4: **5-dataset** statistics. |Train|, |Dev|, |Test| denotes the number of examples in train, dev, test splits respectively. |L| denotes the number of classes for each task.

| Dataset | |Train| | |Dev| | |Test| | |L| |
|---|---|---|---|---|
| MNIST | 51,000 | 9,000 | 10,000 | 10 |
| notMNIST | 15,526 | 2,739 | 459 | 10 |
| Fashion-MNIST | 9,574 | 1,689 | 1,874 | 10 |
| CIFAR10 | 42,500 | 7,500 | 10,000 | 10 |
| SVHN | 62,269 | 10,988 | 26,032 | 10 |

Table 5: **15-dataset-NLP**: Task/Dataset description and statistics. All tasks are either single sentence or sentence pair classification. |Train|, |Dev|, |Test| denotes the number of examples in train, dev, test splits respectively. |L| denotes the number of classes for each tasks.

| Task | Dataset/ Corpus | Domain(s)/ Text source(s) | |Train| | |Dev| | |Test| | |L| | Metrics |
|---|---|---|---|---|---|---|---|
| Linguistic Acceptability | CoLA | Journal articles & books | 7,695 | 856 | 1,043 | 2 | Matthews correlation |
| Boolean Question Answering | BoolQ | Google queries, Wikipedia passages | 8,483 | 944 | 3,270 | 2 | Acc. |
| Sentiment Analysis | SST-2 | Movie reviews | 9,971 | 873 | 872 | 2 | Acc. |
| Paraphrase Detection | QQP | Quora questions | 10,794 | 4,044 | 4,043 | 2 | Acc. & F1 |
| Q & A Categorization | YahooQA | Yahoo! Answers | 13,950 | 4,998 | 4,998 | 10 | Acc. |
| Review Rating Prediction | Yelp | Business reviews | 12,920 | 3,999 | 3,998 | 5 | Acc. |
| Event Factuality | Decomp | FactBank | 10,176 | 4,034 | 3,934 | 2 | Acc. |
| Argument Aspect Detection | AAC | Web documents | 10,893 | 2,025 | 4,980 | 3 | Acc. & F1 |
| Explicit Discourse Marker Prediction | DISCONN8 | Penn Discourse TreeBank | 9,647 | 1,020 | 868 | 8 | Acc. & F1 |
| Question Answering NLI | QNLI | Wikipedia | 9,927 | 5,464 | 5,463 | 2 | Acc. |
| Binary Sentence Order Prediction | RocBSO | Roc story, corpus | 10,000 | 2,400 | 2,400 | 2 | Acc. |
| Natural Language Inference | MNLI | speech, fiction, govt. reports | 11,636 | 4,816 | 4,815 | 3 | Acc. |
| Multi-choice Science QA | SciTAIL | Science exams | 11,145 | 1,305 | 1,304 | 2 | Acc. |
| Implicit Discourse Relation Classification | PDTB2L1 | Penn Discourse TreeBank | 13,046 | 1,183 | 1,046 | 4 | Acc. & F1 |
| Emotion Detection | Emotion | Twitter | 9,600 | 2,000 | 2,000 | 6 | Acc. & F1 |

**5-dataset-NLP** consists of text classification datasets (Zhang et al., 2015) from five diverse domains: (1) news article classification (AGNews, 4-way classification); (2) Yelp sentiment analysis (5-way classification); (3) Amazon sentiment analysis (5-way classification); (4) Wikipedia article classification (DBPedia, 14-way classification); and (5) question and answer topic categorization (YahooQA, 10-way classification). We follow the data processing procedure mentioned in (de Masson d'Autume et al., 2019) and have $115,000$ training examples and $7,600$ test examples per task.

One of the objectives of our work is to study the role of different pre-trained initializations in lifelong learning. To enable this study, we introduce **15-dataset-NLP**, a novel suite of diverse tasks for lifelong learning. It consists of fifteen text classification tasks covering a broad range of domains and data sources. Although there exists a setup with 4 tasks spanning 5 datasets, **5-dataset-NLP** (de Masson d'Autume et al., 2019), we show that our introduced benchmark proves more challenging (see Table 2 and Section 3.3) than the previous setup for the Transformer models (e.g., DistilBERT, BERT, RoBERTa) considered in our study.

**15-dataset-NLP** benchmark consists of single sentence or sentence pair classification tasks. We design our benchmark from existing tasks such that (1) the overall dataset includes various domains, (2) different tasks are (dis)similar to each other, thereby, facilitating both transfer and interference phenomena. All tasks under consideration differ in dataset size (from 8.5k-400k), so for our experiments, we only use between 8.5-14k training examples from each task. Lifelong learning from highly imbalanced data is an interesting problem, and we feel that our introduced benchmark can be used to investigate this problem as well. As our data is gathered from publicly available sources, for some tasks we do not have access to hidden test examples. In such cases, we consider dev examples as test split and sample examples from train split for validation[6]. We describe the tasks below and Table 5 details the evaluation metrics and train/dev/test split sizes for each task.

1. Linguistic acceptability aims at identifying whether the given sequence of words is a grammatical sentence. The Corpus of Linguistic Acceptability (**CoLA**) ((Warstadt et al., 2019) consists of English sentences annotated with their grammatical judgements. The data spans multiple domains, specifically books and journal articles.

2. Boolean QA is a reading comprehension task of answering yes/no questions for a given passage. The Boolean Questions (**BoolQ**) (Clark et al., 2019) dataset consists of short passages with yes/no questions about the passage. The questions are sourced from anonymous Google users and paired up with passages from Wikipedia articles.

3. Sentiment analysis is a binary classification task of identifying the polarity (positive/negative sentiment) of a given text. The Stanford Sentiment Treebank (**SST-2**) (Socher et al., 2013) corpus consists of sentences from Rotten Tomatoes movie reviews annotated with their sentiment.

4. Paraphrase detection aims at identifying whether two sentences are semantically equivalent. The Quora Question Pairs (**QQP**) corpus constitutes of question pairs from **Quora**[7] website annotated for semantic equivalence of question pairs.

5. Q&A categorization is a topic classification task of categorizing question and answer text pairs into existing topics. The Yahoo! Answers Comprehensive Questions and Answers (**YahooQA**) (Zhang et al., 2015) corpus contains data corresponding to the ten largest categories from Yahoo! Webscope program.

6. Review rating prediction is a five-way classification task of predicting the number of stars the user has given in a review given the corresponding text. The **Yelp** (Zhang et al., 2015) dataset contains business reviews obtained from the Yelp Dataset Challenge (2015).

7. Event factuality prediction is the task of determining whether an event described in the text occurred. The factuality annotations from the **Decomp** corpus are recast into an NLI structure and we use the modified dataset from Diverse NLI Collection (Poliak et al., 2018).

8. Argument aspect mining is concerned with the automatic recognition and interpretation of arguments (assessing the stance, source, and supportability for a given topic). The Argument Aspect Corpus (**AAC**) (Stab et al., 2018) has over 25,000 arguments spanning eight topics annotated with three labels (no argument, supporting argument, opposing argument). Stab et al. (2018) collected the data from web documents representing a range of genre and text types, including blogs, editorials, forums, encyclopedia articles.

9. The explicit discourse marker prediction task aims at classifying the discourse markers between sentences. Specifically, words like 'and', 'but', 'because', 'if', 'when', 'also', 'while', 'as' mark the conceptual relationship between sentences (**DISCONN8**) and are

---

[6]We plan to release sampled example indices for replicability of our results
[7]https://www.quora.com/share/First-Quora-Dataset-Release-Question-Pairs

considered as labels for this task as discussed in (Prasad et al., 2019; Kim et al., 2020). We use examples from the Penn Discourse TreeBank 3.0 marked for explicit discourse relationship for our experimentation.

10. Question-answering NLI (**QNLI**) is a task adapted from the SQuAD by converting it into the sentence pair classification task (Wang et al., 2018). QNLI is a binary classification task of detecting whether the context sentence contains the answer to the question.

11. Binary Sentence Ordering (BSO) is a binary classification task to determine the order of two sentences. This task is similar to pre-training objectives considered in recent works. We use Roc Stories (**RocBSO**) (Mostafazadeh et al., 2016) corpus for constructing the dataset for this task.

12. Natural language inference (NLI) is a three-way classification task of predicting whether the premise entails the hypothesis (entailment), contradicts the hypothesis (contradiction), or neither (neutral). The Multi-Genre Natural Language Inference (**MNLI**) (Williams et al., 2018) corpus consists of sentence pairs from different sources (transcribed speech, fiction, and government report) annotated for textual entailment.

13. Multi-choice QA is a reading comprehension task wherein given a passage and question, models need to pick up the right option out of provides ones. Khot et al. (2018) cast the multiple-choice science exam questions into an NLI structure to convert them to the binary classification task. We use the **SciTAIL** (Khot et al., 2018) dataset released by them for our experimentation.

14. Implicit discourse relation classification is a common task of identifying discourse relations between two text spans or arguments. The Penn Discourse TreeBank 3.0 (**PDTB3L1**) (Prasad et al., 2019; Kim et al., 2020) contains a hierarchical annotation scheme (top-level senses, fine-grained level-2 senses) and we use top-level senses comprising of four labels (expansion, comparison, contingency, temporal) for our experimentation.

15. Emotion detection is a classification task of detecting the emotions from a given text snippet. We use **Emotion** dataset (Saravia et al., 2018) which contains Twitter messages with six emotions: anger, fear, joy, love, sadness, and surprise.

### B.1 TASK SEQUENCES

The task sequences for the **Split CIFAR-50** and **Split CIFAR-100** experiments were generated by randomly sampling classes without replacement for each task, similar to Chaudhry et al. (2019). Thus, the sequences were different for every random seed, but since we ran each method with the same 5 seeds, each method was trained and tested on the same 5 sequences.

For **Split YahooQA**, we created 5 tasks by using disjoint groups of consecutive classes (e.g. $\{0, 1\}, \{2, 3\} \dots$). These tasks were than randomly ordered for each task sequence, and each method was trained and tested using the same 5 random sequences.

For **5-dataset**, we randomly select the following dataset orders:

Seq1 SVHN→notMNIST→Fashion-MNIST→CIFAR-10→MNIST

Seq2 SVHN→MNIST→notMNIST→Fashion-MNIST→CIFAR-10

Seq3 CIFAR-10→SVHN→notMNIST→Fashion-MNIST→MNIST

Seq4 notMNIST→Fashion-MNIST→CIFAR-10→SVHN→MNIST

Seq5 CIFAR-10→MNIST→notMNIST→SVHN→Fashion-MNIST

For **5-dataset-NLP**, we randomly select the following dataset orders (first 4 are consistent with (de Masson d'Autume et al., 2019)):

Seq1 Yelp→AGNews→DBPedia→Amazon→YahooQA

Seq2 DBPedia→YahooQA→AGNews→Amazon→Yelp

Seq3 Yelp→YahooQA→Amazon→DBpedia→AGNews

Seq4 AGNews→Yelp→Amazon→YahooQA→DBpedia

Seq5  YahooQA→Yelp→DBPedia→AGNews→Amazon

For **15-dataset-NLP**, we randomly select and use the following 5 orders:

Seq1  Decomp→BoolQ→AAC→Yelp→DISCONN8→SST-2→QQP→YahooQA→QNLI
→RocBSO→MNLI→SciTAIL→CoLA→PDTB2L1→Emotion

Seq2  CoLA→QQP→MNLI→QNLI→Emotion→SST-2→BoolQ→Decomp→AAC→SciTAIL
→RocBSO→Yelp→PDTB2L1→YahooQA→DISCONN8

Seq3  SciTAIL→BoolQ→SST-2→AAC→DISCONN8→YahooQA→QNLI→RocBSO→PDTB2L1
→Emotion→Decomp→MNLI→QQP→CoLA→Yelp

Seq4  DISCONN8→QNLI→CoLA→YahooQA→AAC→SciTAIL→PDTB2L1→Emotion
→Decomp→RocBSO→QQP→Yelp→MNLI→BoolQ→SST-2

Seq5  Emotion→SST-2→RocBSO→YahooQA→AAC→MNLI→CoLA→DISCONN8→QQP
→QNLI→Decomp→PDTB2L1→SciTAIL→Yelp→BoolQ

## C   TASK-SPECIFIC RESULTS

In order to understand the evolution of task-specific performance during continuous training, we visualize the task-specific results in Figures 3 and 4. Specifically, we compare the performance of pre-trained and randomly initialized ResNet-18/ DistilBERT, for the first three tasks in a sequence, across five random task ordering, when evaluated on 5-dataset/ 5-dataset-NLP (diverse tasks). In general, we see that both models start with approximately equal task accuracy (except for CIFAR-10), but pre-trained initialization leads to lesser forgetting than randomly initialized models (consistent with our observation in Figure 1 for Split YahooQA). Moreover, given the heterogeneous nature of the downstream tasks, we see that performance gains (in terms of forgetting) from pre-trained initialization vary across different tasks.

**5-dataset-NLP**   For example, in the case of DBPedia (Figures 3c, 3d, 3o) and AGNews (Figures 3b, 3f, 3j) datasets, we see pre-trained DistilBERT undergoes little to almost no forgetting. One plausible explanation for these results is that both datasets are for the article classification tasks, DBPedia is Wikipedia article classification (14 classes) and AGNews is news article classification (4 classes), and share similar domains with the pre-training corpora (Wikipedia and Books). On the other hand, we see a significant forgetting in the case of Yelp (Figures 3a, 3g, 3k) and Amazon datasets (Figure 3l). Both of these datasets are review sentiment classification tasks (5 classes). We know that the reviews domain (noisy text from Yelp.com and Amazon.com) is less similar with the pre-training corpora (clean text from Wikipedia and Books), and might be one of the reasons behind the drop in performance. Further, note that as we train on the sequence of tasks, we expect to see positive/ negative transfer from related/ unrelated tasks. For example, we see that the performance on Yelp improves significantly after training on Amazon (Figures 3a, 3g, 3n), demonstrating an example of positive transfer from the related task.

**5-dataset**   Here, we report that the forgetting is more severe for SVHN (Figures 4a, 4d, 4h) and CIFAR-10 (Figures 4g, 4l, 4m) as compared to MNIST (Figures 4e, 4n), notMNIST (Figures 4b, 4f, 4i, 4o). Although SVHN and MNIST both are digit recognition tasks, we believe that the realistic nature (house numbers in Google Street View images) of SVHN images makes them more susceptible to forgetting, even in the case of pre-trained ResNet-18 models.

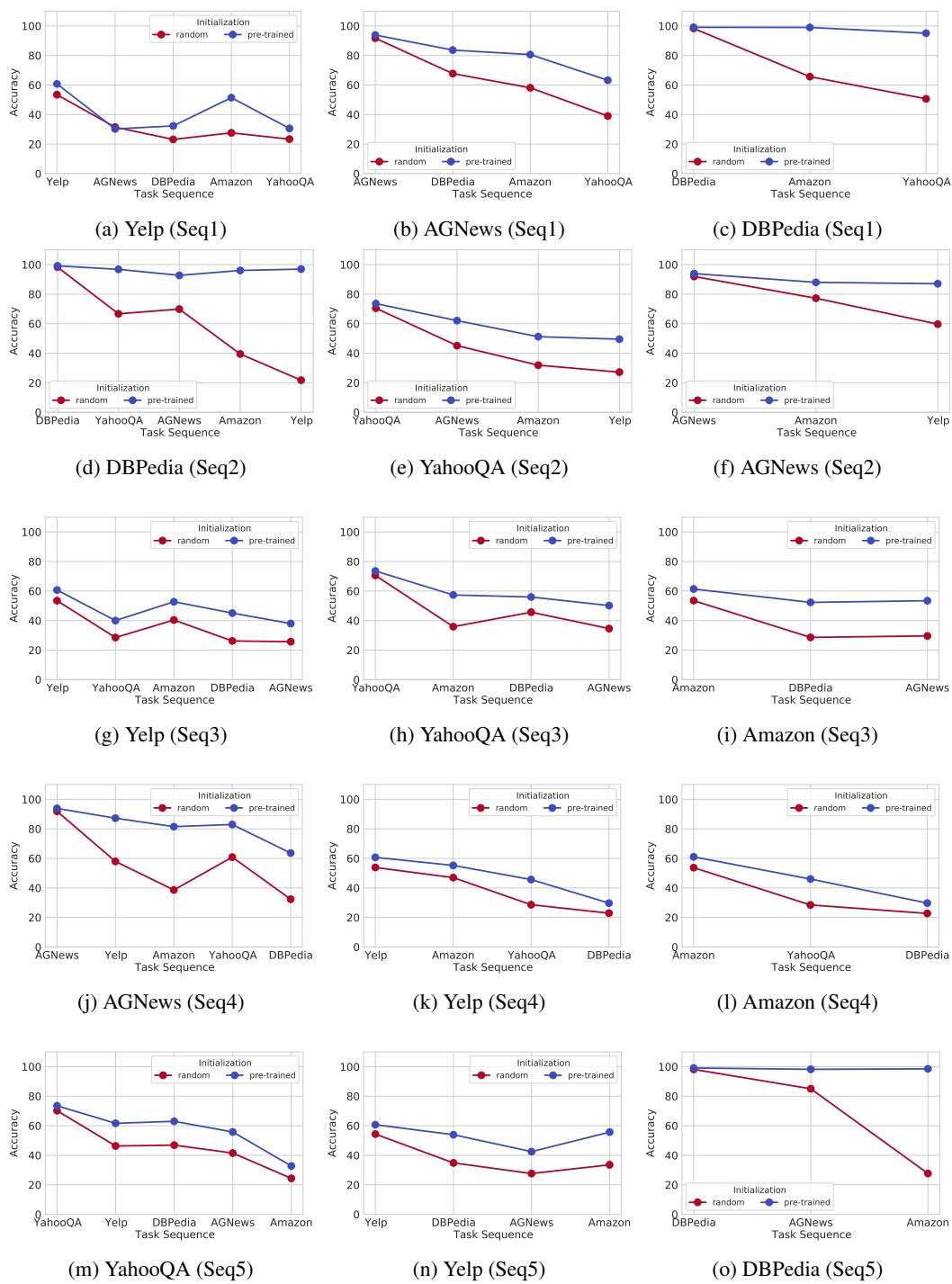

Figure 3: Evolution of task accuracy during sequential training on **5-dataset-NLP**. We compare the performance of pre-trained and randomly initialized models, for first three tasks in a sequence, across five different random task orderings (Seq1,...,Seq5). We see that both models start with approximately equal task accuracy, but pre-trained initialized models undergo lesser forgetting than randomly initialized models.

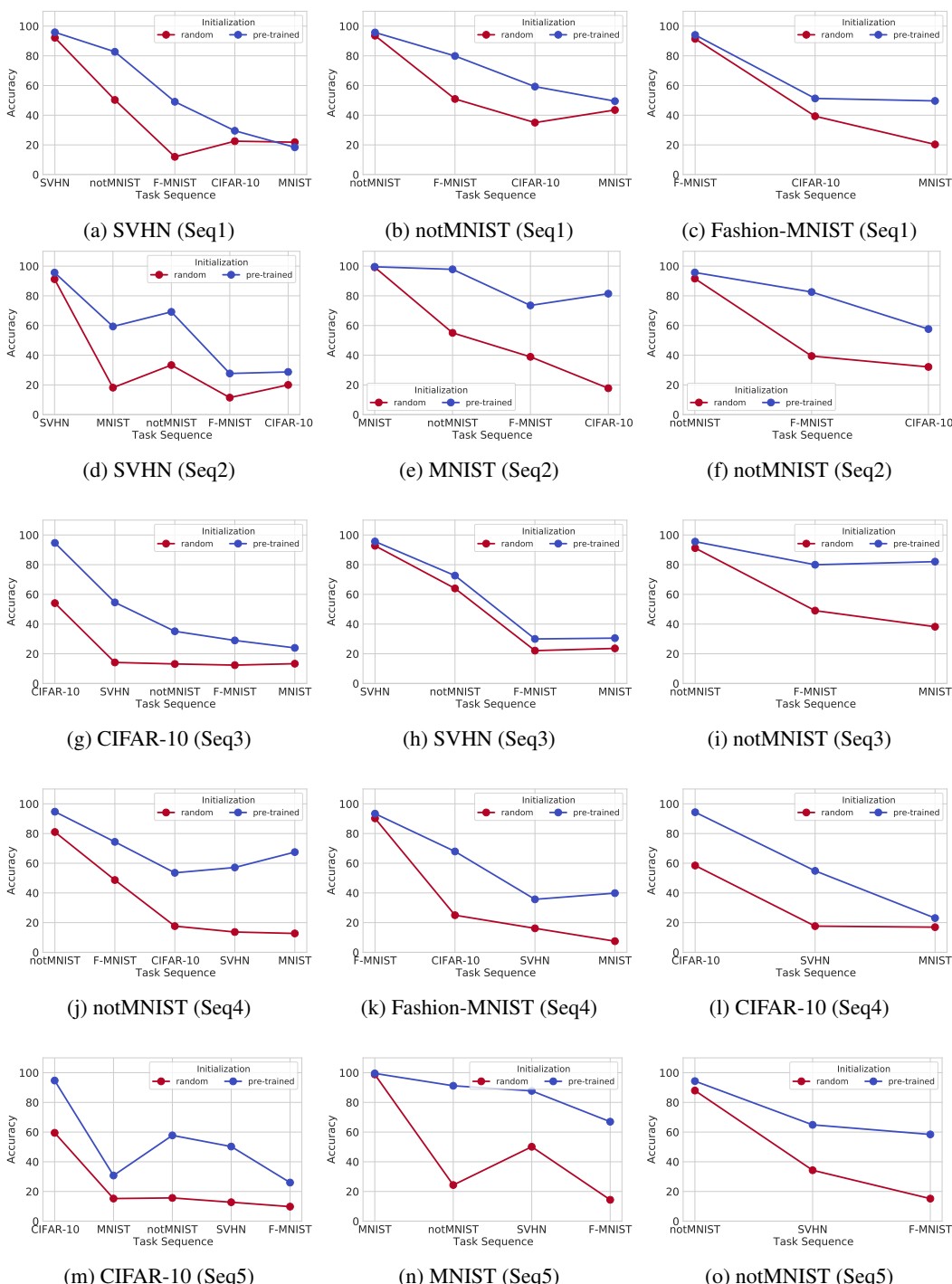

Figure 4: Evolution of task accuracy during sequential training on **5-dataset**. We compare the performance of pre-trained and randomly initialized models, for first three tasks in a sequence, across five different random task orderings (Seq1,...,Seq5). We see that both models start with approximately equal task accuracy (except for CIFAR-10), but pre-trained initialized models undergo lesser forgetting than randomly initialized models.

# D SHARPNESS

The matrix $A \in \mathbb{R}^{n \times p}$ used for projecting the parameters onto a subspace is randomly sampled and then normalized row-wise. Since this matrix is very large, the computation of the pseudo inverse $A^+$ (required for calculating the bounds in Equation 3) is very memory intensive and unstable. Instead, we directly calculate $A^+ x$ by finding the least squares solution to $Ab = x$. To find the maximum referenced in Equation 4, we use the L-BFGS-B algorithm.[8] We set the maximum number of iterations for the algorithm to 10, and to speed up computation, we directly provide the gradients along with the loss to the algorithm, instead of using the default 2-point finite difference gradient estimation.

For ResNet-18 ($n = 11M$), we set $p = 100$. However, for DistilBERT ($n = 66M$) when we set $p = 100$, we notice extremely small values for the sharpness metric. With the increase in the number of parameters, $n$, we should ideally increase random subspace projection dimension $p$. Setting larger $p(> 100)$ values for DistilBERT, however, leads to memory issues relating to allocating space for $A$ and computing the bounds (even with the more efficient method discussed above). So instead of evaluating the sharpness metric in a random manifold, we perform the maximization in the entire space $\mathbb{R}^n$ (basically setting $A = I_n$). According to Keskar et al. (2017), when $\epsilon$ is small enough and $A = I_n$, the sharpness metric in Equation 4 relates to the largest eigenvalue of $\nabla^2 f(x)$. In Table 7, we report sharpness values for DistilBERT on 5-dataset-NLP and Split YahooQA datasets for $\epsilon \in \{5e^{-5}, 1e^{-4}, 5e^{-4}\}$. We see that values in the case of pre-trained models (w/ PT) are lower compared to randomly initialized models (w/o PT), thereby, validating the relative flatness of the task minima in the case of pre-trained models.

Table 6: Average sharpness (lower is flatter) of minima across tasks in a 100 dimensional random subspace. ResNet-18-PT (w/ PT) has lower sharpness in comparison to ResNet-18-R (w/o PT). Pre-training reduces the sharpness of minima for each task in training by an order of magnitude.

|  | $\epsilon = 5 \times 10^{-4}$ | | $\epsilon = 10^{-3}$ | |
| --- | --- | --- | --- | --- |
|  | w/o PT | w/ PT | w/o PT | w/ PT |
| Split CIFAR-100 | 2.26($\pm$0.58) | 0.10($\pm$0.02) | 5.89($\pm$1.33) | 0.24($\pm$0.06) |

Table 7: Average sharpness (lower is flatter) of tasks minima. DistilBERT-PT (w/ PT) reduces the sharpness in comparison to DistilBERT-R (w/o PT).

|  | $\epsilon = 5 \times 10^{-5}$ | | $\epsilon = 10^{-4}$ | | $\epsilon = 5 \times 10^{-4}$ | |
| --- | --- | --- | --- | --- | --- | --- |
|  | w/o PT | w/ PT | w/o PT | w/ PT | w/o PT | w/ PT |
| 5-dataset-NLP | $32.67 \pm 1.17$ | $28.27 \pm 1.19$ | $213.61 \pm 11.46$ | $128.97 \pm 10.49$ | $596.82 \pm 13.70$ | $552.09 \pm 17.28$ |
| Split YahooQA | $10.41 \pm 0.39$ | $8.77 \pm 0.44$ | $53.23 \pm 7.02$ | $43.03 \pm 4.21$ | $545.06 \pm 6.40$ | $422.85 \pm 44.31$ |

# E LOSS LANDSCAPE

## E.1 LINEAR MODEL INTERPOLATION

Ideally, to reduce forgetting, as the model sequentially trains on tasks, its loss on previous tasks should change minimally. This would be satisfied if the loss surface between model checkpoints were flat. To visualize this, we linearly interpolate between $w_1$ and the other checkpoints of the model, tracking the test loss on task 1 and task 2. This can be interpreted as viewing a slice of the contour plots shown in Figure 2 along the line that connects $w_1$ to each checkpoint. The results are shown in Figure 5, with first row for task 1 and second row for task 2. The pre-trained plots are shown in hues of blue, and the random plots are shown in hues of red. These plots show that the pre-trained initialized models experience a much more gradual increase in the loss compared to the randomly initialized models, even when interpolating to checkpoints created after training on several tasks.

---

[8]We used the implementation provided by scipy at https://docs.scipy.org/doc/scipy/reference/optimize.minimize-lbfgsb.html

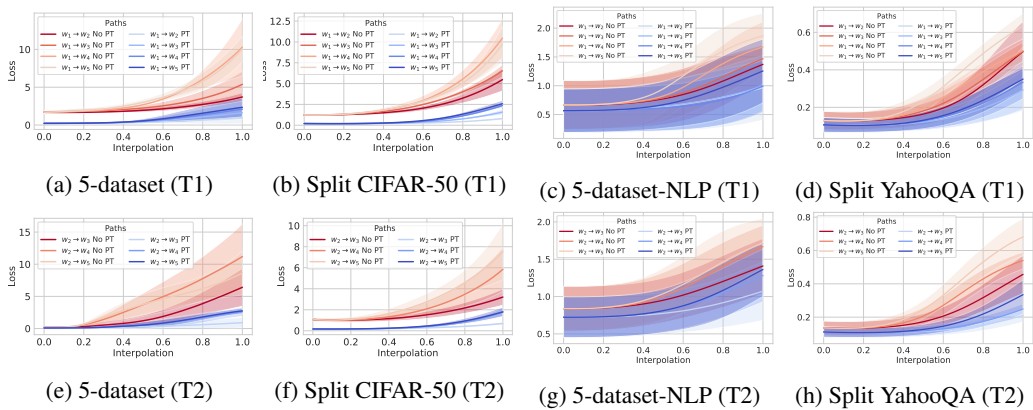

(a) 5-dataset (T1)    (b) Split CIFAR-50 (T1)    (c) 5-dataset-NLP (T1)    (d) Split YahooQA (T1)

(e) 5-dataset (T2)    (f) Split CIFAR-50 (T2)    (g) 5-dataset-NLP (T2)    (h) Split YahooQA (T2)

Figure 5: Loss interpolation plots for each dataset. Blue is pre-trained models, red is randomly initialized models. We interpolate between the checkpoint after **task 1** (T1)/ **task 2** (T2) to the checkpoint after every other task, tracking the loss in the process. In general, the loss landscape is flatter along these paths for pre-trained initialized models compared to randomly initialized models.

Moreover, these results hold for task 2 as well, thereby verifying that pre-trained initialized models lead to flatter task minima for subsequent tasks.

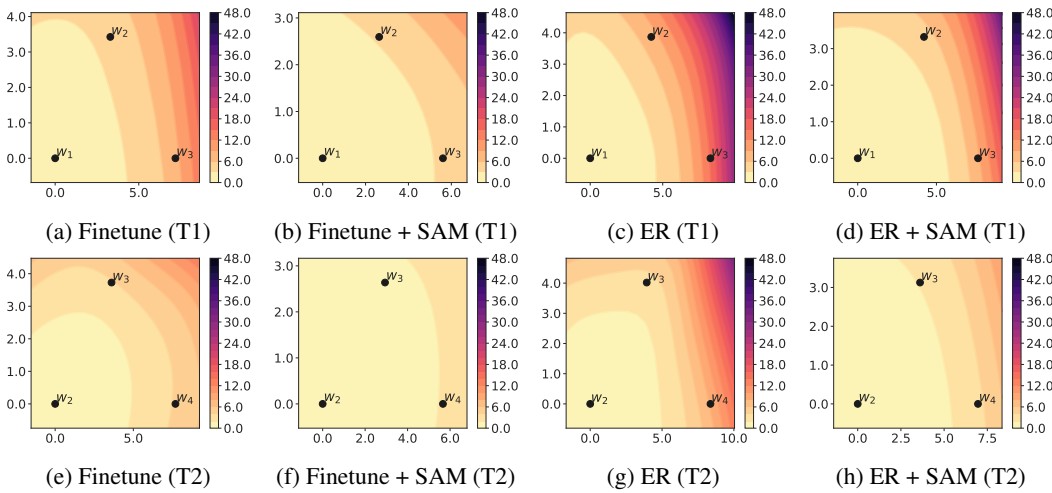

(a) Finetune (T1)    (b) Finetune + SAM (T1)    (c) ER (T1)    (d) ER + SAM (T1)

(e) Finetune (T2)    (f) Finetune + SAM (T2)    (g) ER (T2)    (h) ER + SAM (T2)

Figure 6: Loss contours for **task 1** (T1) and **task 2** (T2) of **Split CIFAR-50**. The top row visualizes loss contours for task 1 where $\mathbf{w}_1, \mathbf{w}_2, \mathbf{w}_3$ are minima obtained after sequential training on tasks 1, 2, and 3, respectively. Similarly, the bottom row visualizes loss contours for task 2 after sequential training on tasks 2, 3, and 4. All of the above models start with random weights. SAM (Finetune + SAM, ER + SAM) leads to wide task minima compared to Finetune and ER methods.

## E.2 LOSS CONTOURS

In this section we present loss contours for task 1/ task 2 for all task sequences (refer to Section B.1 for more details) for **5-dataset-NLP**, **Split YahooQA**, **Split CIFAR-50**, and **5-dataset**. In line with our observation from the sharpness and linear model interpolation analyses, pre-trained initialized models lead to flatter task minima for subsequent tasks as well.

### E.2.1 LOSS CONTOURS: SAM AND ER

We plot loss contours for Task 1/ Task 2 of Split CIFAR-50 (Figure 6) and 5-dataset (Figure 7), under continual training from randomly initialized weights, and compare them across four different methods: Finetune, Finetune + SAM, ER, and ER + SAM. We notice that SAM (Finetune + SAM and ER + SAM) leads to wide task minima (Task 1/ Task 2) across both datasets as compared to Finetune and ER methods. Moreover, from Table 1, we see that for Split CIFAR-50, Finetune + SAM (14.95), ER + SAM (16.88) undergoes lesser forgetting than Finetune (23.68) and ER (20.63) methods. These results convincingly demonstrate the effectiveness of SAM when used with vanilla SGD and/ or ER methods. Similarly, we see that for 5-dataset, ER + SAM (27.32) undergoes lesser forgetting than Finetune + SAM (40.92), which in turn improves over Finetune (51.51). Next, we compare the loss contours between Finetune and ER methods and do not notice any stark difference in terms of flatness. However, in the presence of SAM, qualitatively we see that ER + SAM (Figures 7d, 7h) leads to a flat loss basin in comparison to Finetune + SAM (Figures 7b, 7f).

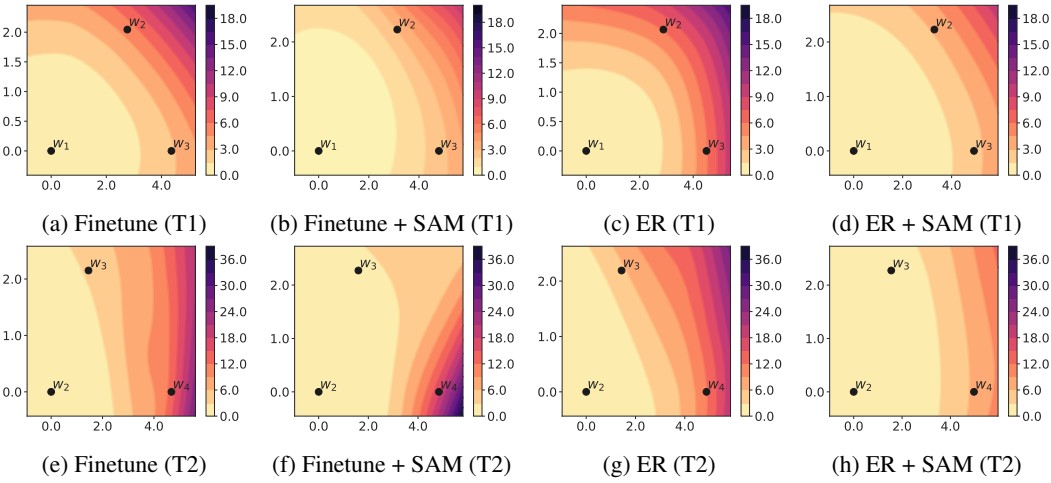

Figure 7: Loss contours for **SVHN** (T1) and **MNIST** (T2) of **5-dataset**. The top row visualizes loss contours for SVHN where $\mathbf{w}_1$, $\mathbf{w}_2$, $\mathbf{w}_3$ are minima obtained after sequential training on SVHN, MNIST, and nonMNIST, respectively. Similarly, the bottom row visualizes loss contours for MNIST where $\mathbf{w}_2$, $\mathbf{w}_3$, $\mathbf{w}_4$ are minima obtained after sequential training on tasks MNIST, nonMNIST, and Fashion-MNIST. All of the above models start with random weights. SAM (Finetune + SAM, ER + SAM) leads to wide task minima compared to Finetune and ER methods.

### E.2.2 LOSS CONTOURS: 5-DATASET-NLP

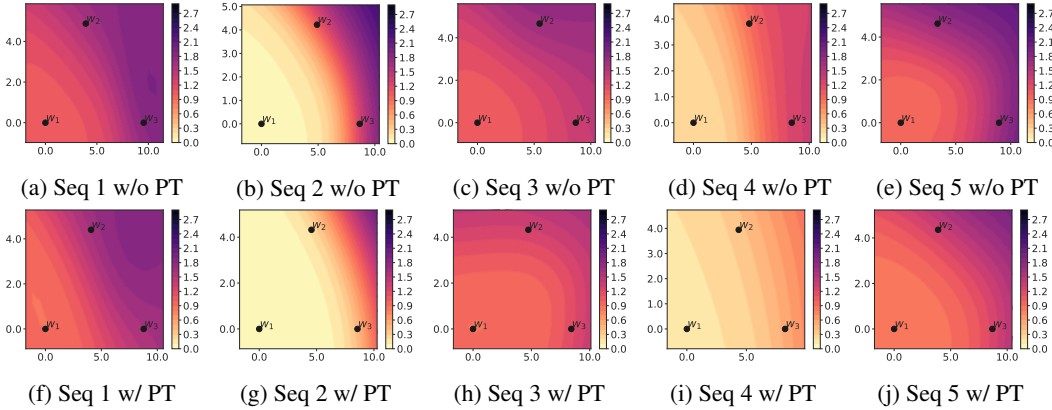

Figure 8: Loss contours for **Task 1** on 5 task sequences of **5-dataset-NLP**. Each contour shows the location of the model parameters after training sequentially on **Task 1 ($w_1$), Task 2 ($w_2$), Task 3 ($w_3$)**. The top row shows contours for randomly initialized models (w/o PT) and the bottom row shows contours for pre-trained initialized models (w/ PT).

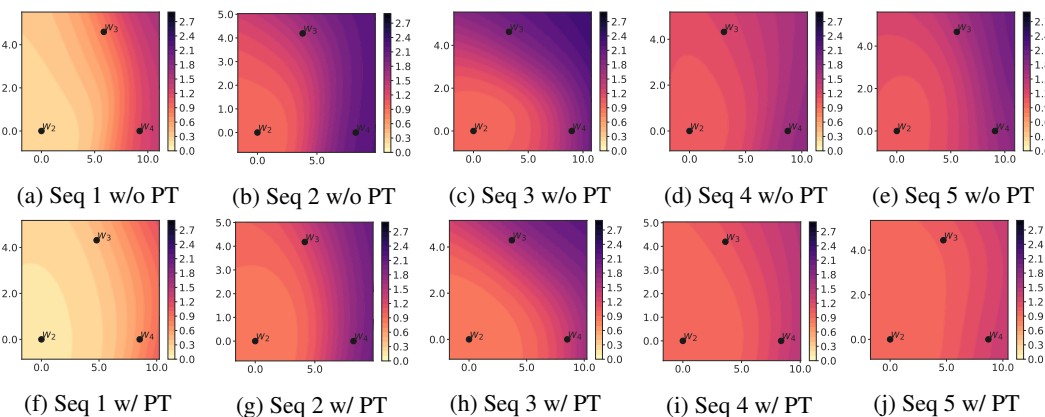

Figure 9: Loss contours for **Task 2** on 5 task sequences of **5-dataset-NLP**. Each contour shows the location of the model parameters after training sequentially on **Task 2 ($w_2$), Task 3 ($w_3$), Task 4 ($w_4$)**. The top row shows contours for randomly initialized models (w/o PT) and the bottom row shows contours for pre-trained initialized models (w/ PT).

### E.2.3 Loss Contours: Split YahooQA

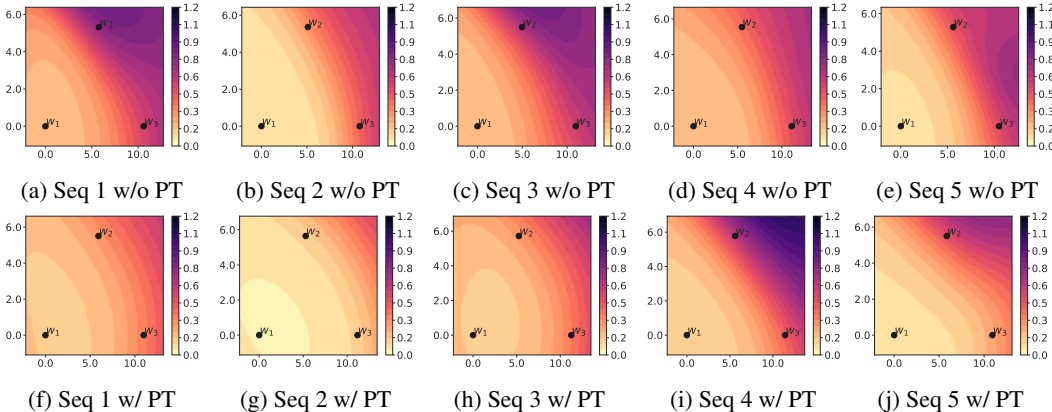

Figure 10: Loss contours for **Task 1** on 5 task sequences of **Split YahooQA**. Each contour shows the location of the model parameters after training sequentially on **Task 1 ($w_1$), Task 2 ($w_2$), Task 3 ($w_3$)**. The top row shows contours for randomly initialized models (w/o PT) and the bottom row shows contours for pre-trained initialized models (w/ PT).

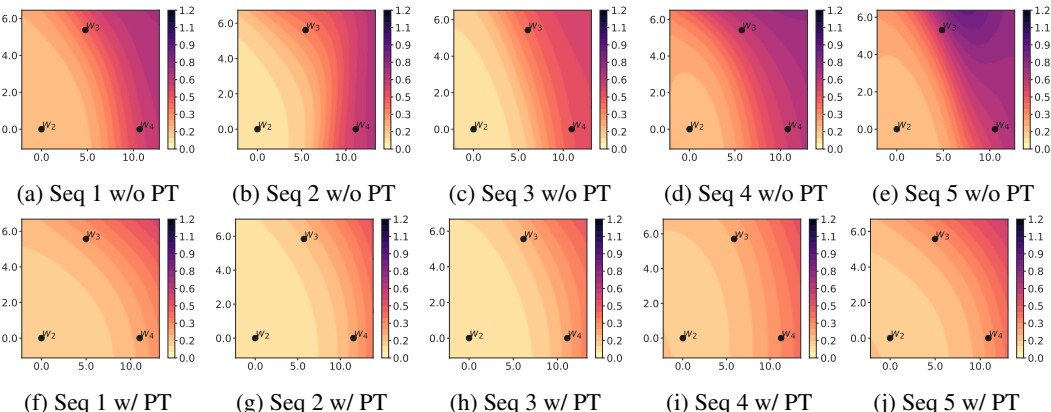

Figure 11: Loss contours for **Task 2** on 5 task sequences of **Split YahooQA**. Each contour shows the location of the model parameters after training sequentially on **Task 2 ($w_2$), Task 3 ($w_3$), Task 4 ($w_4$)**. The top row shows contours for randomly initialized models (w/o PT) and the bottom row shows contours for pre-trained initialized models (w/ PT).

### E.2.4 LOSS CONTOURS: SPLIT CIFAR-50

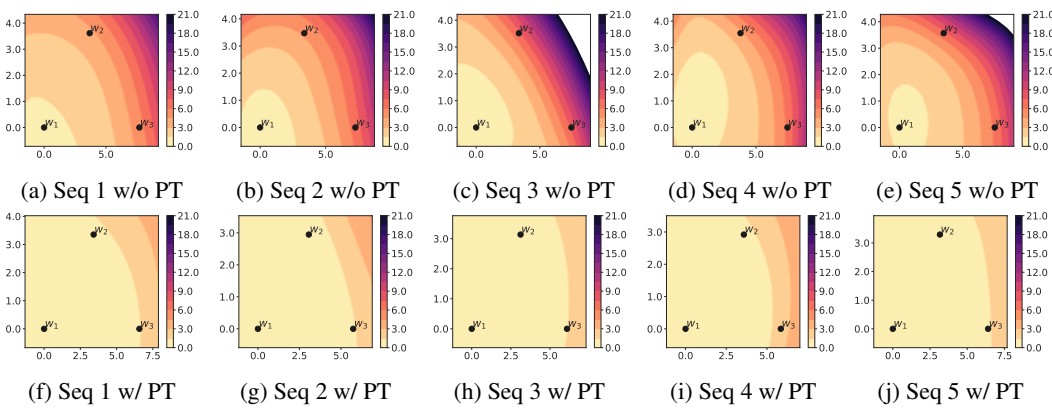

Figure 12: Loss contours for **Task 1** on 5 task sequences of **Split CIFAR-50** with 5 epochs of training on each task. Each contour shows the location of the model parameters after training sequentially on **Task 1 ($w_1$), Task 2 ($w_2$), and Task 3 ($w_3$)**. The top row shows contours for randomly initialized models (w/o PT) and the bottom row shows contours for pre-trained initialized models (w/ PT).

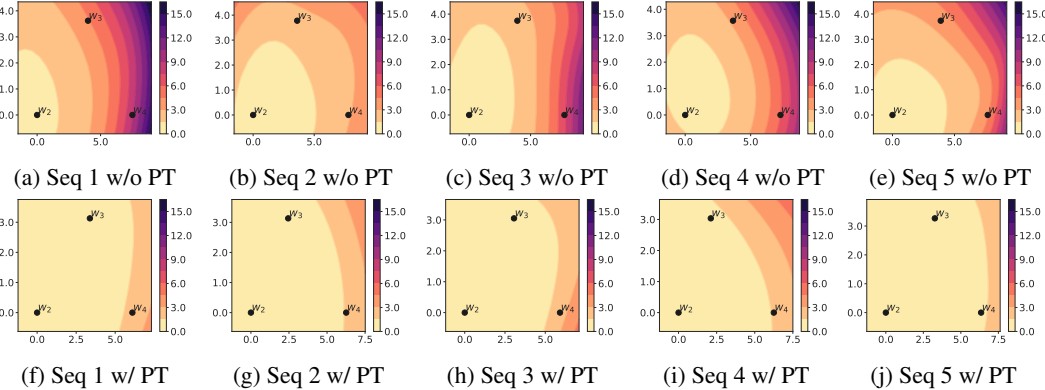

Figure 13: Loss contours for **Task 2** on 5 task sequences of **Split CIFAR-50** with 5 epochs of training on each task. Each contour shows the location of the model parameters after training sequentially on **Task 2 ($w_2$), Task 3 ($w_3$), and Task 4 ($w_4$)**. The top row shows contours for randomly initialized models (w/o PT) and the bottom row shows contours for pre-trained initialized models (w/ PT).

### E.2.5 LOSS CONTOURS: 5-DATASET

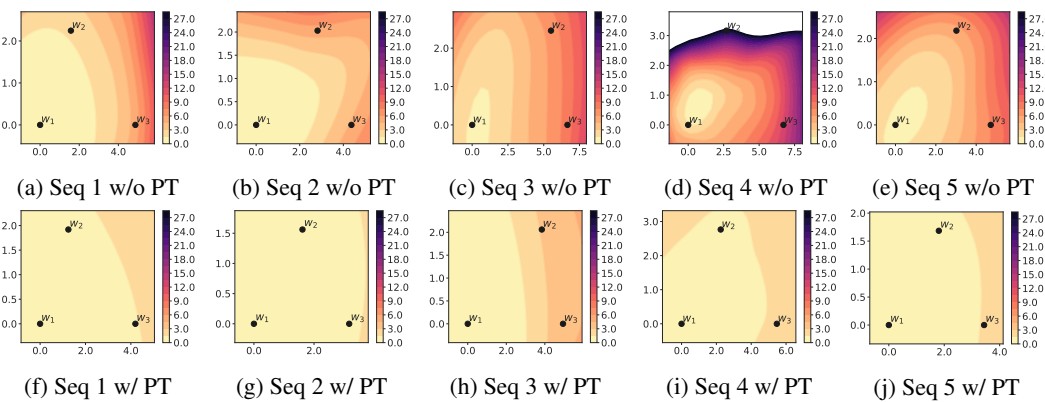

Figure 14: Loss contours for **Task 1** on 5 task sequences of **5-dataset** with 5 epochs of training on each task. Each contour shows the location of the model parameters after training sequentially on **Task 1 ($w_1$), Task 2 ($w_2$), and Task 3 ($w_3$)**. The top row shows contours for randomly initialized models (w/o PT) and the bottom row shows contours for pre-trained initialized models (w/ PT).

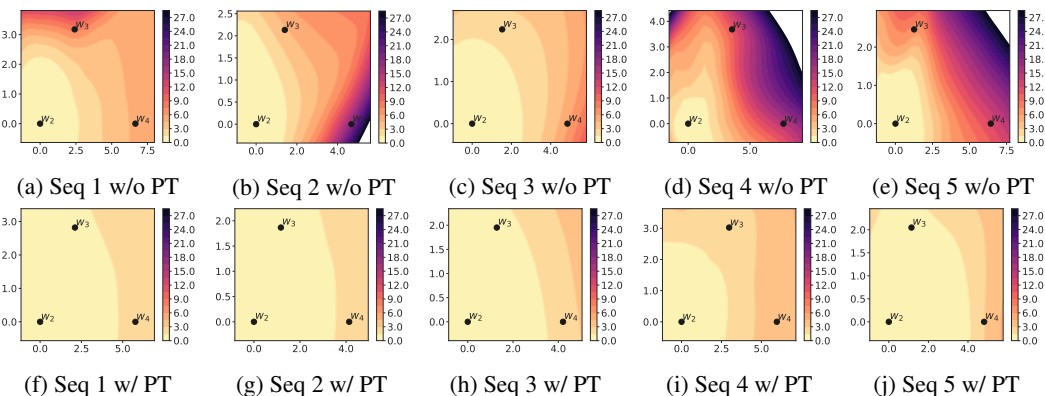

Figure 15: Loss contours for **Task 2** on 5 task sequences of **5-dataset** with 5 epochs of training on each task. Each contour shows the location of the model parameters after training sequentially on **Task 2 ($w_2$), Task 3 ($w_3$), and Task 4 ($w_4$)**. The top row shows contours for randomly initialized models (w/o PT) and the bottom row shows contours for pre-trained initialized models (w/ PT).

## F  ADDITIONAL RELATED WORK

**Transfer learning** from generic pre-trained models has enabled significant recent progress in ML (Zhuang et al., 2021). This trend started in the CV field with the ImageNet dataset (Deng et al., 2009). Transfer learning in NLP has witnessed its own "ImageNet revolution" where large models pre-trained on self-supervised tasks have shown impressive results across many language understanding tasks (Peters et al., 2018; Howard & Ruder, 2018; Radford et al., 2018; Devlin et al., 2019; Raffel et al., 2019; Liu et al., 2019).

**Lifelong learning** approaches focus on the idea of mitigating the catastrophic forgetting phenomenon and can be categorized into three groups: (1) *Regularization-based* approaches that augment the loss function with extra penalty terms preventing important parameters learned on previous tasks from significantly deviating while training on the new task (Kirkpatrick et al., 2017; Zenke et al., 2017); (2) *memory-based* approaches that augment the model with episodic memory for sparse experience

replay of previous task examples (Lopez-Paz & Ranzato, 2017; Chaudhry et al., 2018; Wang et al., 2020) (3) *network expansion-based* approaches that dynamically expand the network based upon new tasks (Rusu et al., 2016; Aljundi et al., 2017; Sodhani et al., 2020). We analyze regularization-based and memory-based approaches in this work.

**Meta-learning** involves creating models that learn to learn over time. Several works have applied meta-learning approaches to the task of lifelong learning (Riemer et al., 2019; Finn et al., 2019; Javed & White, 2019; Wang et al., 2020). Caccia et al. (2020) propose a two-phase continual learning scenario where the first phase is pre-training (using MAML (Finn et al., 2017)) and the second phase involves continual deployment with task revisiting. They make the point that in many scenarios (Lomonaco et al., 2019), it would be unrealistic to deploy agents with no pre-training in a lifelong learning setting. Whereas some of these works do use pre-trained initializations for their models, many do not, and none have extensively studied the effect of pre-training on lifelong learning.

