# OpenReview forum: "An Empirical Investigation of the Role of Pre-training in Lifelong Learning"
_ICLR.cc/2022/Conference — ICLR 2022 Submitted_

### Official Review · Reviewer_xCMa · 2021-11-02

**Correctness:** 4
**Technical Novelty And Significance:** 2
**Empirical Novelty And Significance:** 2
**Recommendation:** 6
**Confidence:** 4

**Main Review:**

Strong points:
 - This paper takes a different and interesting approach to tackle one of the most important issues of lifelong learning i.e., catastrophic forgetting. As pre-training of large models on large diverse datasets has become a norm, this approach looks more practical to me than fine-grained changes in rehearsal-based and regularization based approaches.
 - The main hypothesis that pre-training helps in reducing catastrophic forgetting over a sequence of datasets/tasks has been supported theoretically and empirically with diverse quantitative results and qualitative analyses.
 - To strongly demonstrate the validity of main hypothesis, this paper provides comprehensive experiments, including both quantitative results and qualitative analysis: (1) using prominent approaches of lifelong learning in recent years consisting of rehearsal based (A-GEM, ER) and regularization based (EWC) approaches, (2) for Computer Vision (CV) and Natural Language Processing (NLP) applications, (3) on homogeneous (Split CIFAR-100, Split Cifar-50, Split YahooQA) and diverse (5-dataset, 5-dataset-NLP, 15-dataset-NLP) sequential datasets.
 - For example, it has been demonstrated that a simple fine-tuning (without regularization or rehearsal) of a pre-trained model on a sequence of diverse tasks (5-dataset-NLP) undergoes significantly less forgetting than rehearsal-based ER method with randomly initialized weights, which begs the question that if instead of specifically focusing on forgetting aspect of lifelong learning would it be more useful to learn generic features on a larger scale?
 - In case of diverse sequential datasets (5-dataset, 5-dataset-NLP) simple fine-tuning of a pre-trained model performs worse than ER method which shows that pre-trained models are susceptible to forgetting when learning on diverse tasks. However, larger models and models pre-trained with diverse pre-trained corpora undergoes less forgetting which shows that diversity of pre-trained corpus is a relevant factor during lifelong learning.
 - Experiments have been performed with multiple different sequences of tasks in each dataset to test robustness of the proposed approach.
 - Loss contour plots and Sharpness metric evaluations further validates the hypothesis that pre-trained models have bias for wider loss basins for fine-tuning tasks.
 - It has been shown that in addition to pre-trained weight initialization, it further helps in reducing forgetting if sharpness of the loss basin is taken into account as an additional training objective via Sharpness-Aware Minimization (SAM) training procedure. Experimental results confirm that SAM eases catastrophic forgetting and improves the performance of underlying model whether it is "fine-tuning of a pre-trained model" or "episodic replay (ER)".

Weak points:
 - Following the setup of experiments of ER method with and without SAM training procedure, it would be interesting to see the experiments with model initialized with pre-trained weights in ER method to see if fine-tuning of pre-trained weights and rehearsal of examples from previous tasks complement each other or not.
 - To see the effects of SAM optimization procedure qualitatively, I would recommend to plot loss contours for Task 1 (as shown in Figure 2) for Finetune + SAM models.
 - Also, it would be interesting to see loss contours for Task 1 using models trained with ER method with randomly initialized weights to compare Finetune and ER methods qualitatively.

**Summary Of The Paper:**

This paper explores the catastrophic forgetting in lifelong learning from the perspective of model initialization. It further shows that large pre-trained models have an implicit bias towards wider loss basin when fine-tuning on sequential tasks which ultimately results in less forgetting of tasks when training on subsequent tasks. Furthermore, by explicitly seeking wider loss basins during lifelong learning training as an additional optimization objective further reduces the forgetting regardless of the lifelong learning algorithm and base model used.

**Summary Of The Review:**

Although there is no novel algorithm or model introduced in this paper, but the direction it explores to minimize catastrophic forgetting in lifelong learning methods via pre-trained weight initialization is interesting and practical as large pre-trained models have become a norm because of the wide range of diverse knowledge they can capture. Even though the Finetune method is not the best performing method in all of the experiments (however it outperforms in most of the experiments), the key conclusion I can take form this paper is that instead of completely focusing on improving upon fine-grained details of forgetting aspect of lifelong learning, the focus should also be on learning more general representations as they appear to result in robust lifelong learning.

---

> ### Author Response · Authors · 2021-11-20
> **Response to Reviewer xCMa (Part 1/1)**
>
> We thank the reviewer for the positive review and valuable suggestions! We highly appreciate the reviewer’s remarks that: (i) our work takes a different and interesting approach to tackle catastrophic forgetting rather than fine-grained changes in existing approaches, and (ii) our claims are supported theoretically and empirically with comprehensive experiments, including both qualitative and quantitative analyses.
>
> **[ER w/wo Rehearsal]**
>
> In Tables 1 and 2, we already report results for Finetune and ER methods with and without SAM optimization procedure, both for randomly initialized and pre-trained models. For experiments with models initialized with pre-trained weights, we observe that ER performs superior to Finetune, across both CV and NLP domains. Note that if we remove the episodic rehearsal component from ER, then it boils down to the Finetune method. Therefore, whatever additional gains (compared to Finetune) we see with ER, can be attributed primarily to a rehearsal of examples from previous tasks. Surprisingly, for discrete NLP inputs, even a single example per class per the previous task significantly alleviates the forgetting.
>
> **[Loss contours: SAM and ER]**
>
> In order to understand the effectiveness of the SAM, we compute the sharpness metric for Finetune and Finetune + SAM methods. We have updated Table 3 with sharpness metrics for 5-dataset and Split CIFAR-50. We see that the SAM significantly reduces the sharpness in the case of randomly initialized models. Concretely, on the 5-dataset, we see that sharpness value (for $\epsilon=5$ x $10^{-4}$) decreases from 2.07 (Finetune) to 0.69 (Finetune + SAM). Similarly, on the Split CIFAR-50, we see a drop from 2.26 (Finetune) to 0.73 (Fine + SAM). These results validate that SAM indeed leads to flat minima, therefore, explaining the superior performance (in terms of average accuracy and forgetting) of SAM optimization procedure over vanilla SGD fine-tuning.
>
> **As suggested by the reviewer, we also plot loss contours for Task 1/ Task 2 of Split CIFAR-50 ( Figure 6, Appendix section E.2.1) and 5-dataset (Figure 7, Appendix section E.2.1)**, under continual training from randomly initialized weights, and compare them across four different methods: **Finetune, Finetune + SAM, ER, and ER + SAM**. We notice that SAM (Finetune + SAM and ER + SAM) leads to wide task minima (Task 1/ Task 2) across both datasets as compared to Finetune and ER methods. Moreover, from Table 1, we see that for Split CIFAR-50, Finetune + SAM (14.95), ER + SAM (16.88) undergoes lesser forgetting than Finetune (23.68) and ER (20.63) methods. These results convincingly demonstrate the effectiveness of SAM when used with vanilla SGD and/ or ER methods. Similarly, from Table 1, we see that for 5-dataset, ER + SAM (27.32) undergoes lesser forgetting than Finetune + SAM (40.92), which in turn improves over Finetune (51.51).
>
> Next, we compare the loss contours between Finetune and ER methods and do not notice any stark difference in terms of flatness. However, in the presence of SAM, qualitatively we see that ER + SAM (Figures 7(d), 7(h)) leads to a flat loss basin in comparison to Finetune + SAM (Figures 7(b), 7(f)).
>
>
> *We thank the reviewer for suggesting us the qualitative analysis. We have updated the paper with these qualitative results.*

---

### Official Review · Reviewer_UX5R · 2021-11-02

**Correctness:** 3
**Technical Novelty And Significance:** 2
**Empirical Novelty And Significance:** 2
**Recommendation:** 5
**Confidence:** 4

**Main Review:**

Empirical analysis papers like this one, aiming to identify and understand best training practices, are sorely needed in machine learning. Authors have made substantial efforts in building multiple datasets, comparing different settings, and trying to understand what their findings mean. The proposed loss constraint solution appears to work well, and improves overall model performance.

Despite the large amount of work carried out, I have several concerns about the proposed work and its conclusions.
Firstly, the main conclusions of the different experiments are not providing significant novel insights. It is well known that model pre-training provides significant performance boosts and robust transfer training solutions. It is expected that pre-training would yield superior performance in datasets somewhat similar to the pre-training dataset. Furthermore, comparing model weights and loss landscape between a robust pre-trained model and a randomly initialised, potentially not converged model does not seem particularly fair. The large gap between learning accuracies in certain settings highlights this issue. The flat loss landscape is a more interesting, yet not novel, conclusion as acknowledged in section 6.

One key limitation is that, besides analysing forgetting, the experiments proposed here provide little insights with regards, specifically, to the task incremental setting. For example, task dissimilarity and ordering could have an impact on how beneficial pre-training is. What happens if the first task is highly different from the pre-training dataset?  What is  the impact of the choice of the pre-training dataset? This was briefly discussed when comparing different BERT models, but the main conclusion was that larger and more diverse pre-training datasets are better, which is also a well known result. Authors had a great opportunity to analyse the impact of the dissimilarity between the pre-training and task specific datasets on forgetting with their 5(+)-datasets settings. It would have been very interesting to study the impact of ordering tasks differently depending on their distance to the pre-training dataset.

Furthermore, reported results only provide overall accuracy and forgetting. Task specific results would provide much more insights, especially when dealing with heterogeneous tasks. Are observed differences consistent across tasks? Are there tasks where pre-training is detrimental, or less beneficial?

The presentation of the paper could be improved greatly. There is a lot of content, which makes the paper extremely condensed with a lot of essential information relegated to supplementary material such as the discussion on related work. The issue is also reflected in the presentation of the results. Different experiments are reported in the same table, which is referenced multiple times at different points in the paper. This leads the reader to wonder where to look in the table, and what the SAM method is, not knowing that it will be introduced at the end of the paper. This creates confusion.

In addition, authors provide very little information on datasets. It is not stated in the main paper why these specific datasets are chosen, whether their order was curated, and how tasks were constructed. Again, a lot of the essential information is relegated to the supplementary material when this constitutes a key contribution.

Similarly, while there exist a large set of methods aiming to address the task incremental learning problem, authors selected 4 without clear justification. Why were these methods selected over others?

Minor comments:

How were semantically similar classes between ImageNet and CIFAR determined?

Please double check citation formats. Some were not accurately formatted in the text.

Please add line number to facilitate the review process and point out typos.


**Summary Of The Paper:**

The authors present a comparative study on the impact of using pre-trained models on task incremental learning problems. They split computer vision and NLP datasets into multiple tasks and evaluate performance between pre-trained and randomly initialised models. They conclude that task incremental learning benefits from pre-training due to the flatter loss landscapes, and propose to use an optimisation constraint to learn better models.

**Summary Of The Review:**

In summary, authors provide a study with a lot of potential, but unfortunately the paper in its current state does not provide substantially novel conclusions or innovations. While the proposed experiment are substantial and do provide additional confirmations, main conclusions are mostly well known (pre-training achieves better performance, larger datasets and models are better), or already discussed in the lifelong learning community (flat loss landscape).  The proposed SAM optimisation constraint is a direct application of another work, and its interaction with different continual learning methods, which often rely on optimisation constraints, is not discussed.
Finally, the paper is presenting a lot of content which is severely hurting presentation and readability, with essential elements relegated to supplementary material or missing. I would recommend refocusing the work or submitting in settings where page limit is not a concern.

---

> ### Author Response · Authors · 2021-11-20
> **Response to Reviewer UX5R (Part 3/3)**
>
> **[Task-specific results]**
>
> As discussed before, one of the desiderata of a lifelong learning method is to be robust to different task orderings. Therefore, we run all of our experiments with five random task orderings and report average accuracy, forgetting (backward transfer), and learning accuracy metrics. These metrics help us to compare different methods in terms of overall performance and forgetting. **In order to understand the evolution of task-specific performance during continuous training, we visualize the task-specific results in Figures 3 and 4 (Appendix C)**. Specifically, we compare the performance of pre-trained and randomly initialized ResNet-18/ DistilBERT, for the first three tasks in a sequence, across five random task ordering, when evaluated on 5-dataset/ 5-dataset-NLP (diverse tasks). In general, we see that both models start with approximately equal task accuracy (except for CIFAR-10), but pre-trained initialization leads to lesser forgetting than randomly initialized models (consistent with our observation in Figure 1 for Split YahooQA). Moreover, given the heterogeneous nature of the downstream tasks, we see that performance gains (in terms of forgetting) from pre-trained initialization vary across different tasks.
>
> **5-dataset-NLP** For example, in the case of DBPedia (Figures 3(c), 3(d), 3(o)) and AGNews (Figures 3(b), 3(f), 3(j)) datasets, we see pre-trained DistilBERT undergoes little to almost no forgetting. One plausible explanation for these results is that both datasets are for the article classification tasks, DBPedia is Wikipedia article classification (14 classes) and AGNews is news article classification (4 classes), and share similar domains with the pre-training corpora (Wikipedia and Books). On the other hand, we see a significant forgetting in the case of Yelp (Figures 3(a), 3(g), 3(k)), and Amazon datasets (Figure 3(l)). Both of these datasets are review sentiment classification tasks (5 classes). We know that the domain of the review (noisy text from Yelp.com and Amazon.com) is less similar to the pre-training corpora (clean text from Wikipedia and Books), and might be one of the reasons behind the drop in performance. Further, note that as we train on the sequence of tasks, we expect to see positive/ negative transfer from related/ unrelated tasks. For example, we see that the performance on Yelp improves significantly after training on Amazon (Figures 3(a), 3(g), 3(n)), demonstrating an example of positive transfer from the related task.
>
> **5-dataset** Here, we report that the forgetting is more severe for SVHN (Figures 4(a), 4(d), 4(h)) and CIFAR-10 (Figures 4(g), 4(l), 4(m)) as compared to MNIST (Figures 4(e), 4(n)), notMNIST (Figures 4(f), 4(i), 4(j), 4(o)). Although SVHN and MNIST both are digit recognition tasks, we believe that the realistic nature (house numbers in Google Street View images) of SVHN images makes them more susceptible to forgetting, even in the case of pre-trained ResNet-18 models.
>
> We believe that the above discussion based upon task-specific results unlocks interesting questions for future studies (similar in line to our discussion under **[Task similarity/dissimilarity]** response section). We thank the reviewer for the insightful questions as they are truly helpful to spin up follow-up works.
>
> **[Clarification question]**
>
> We use the publicly available [4] two-level class hierarchies for ImageNet, where semantically and visually similar labels are grouped under one super-category. We iterate over all CIFAR-100 labels and drop the complete super-category from ImageNet corresponding to each of these labels. For example, CIFAR-100 contains a *castle* class and we have a *building* super-category in ImageNet that contains *castle, palace, monastery, church, etc.*. We remove all building-related labels from our pre-training dataset. In total, we remove 267 classes and pre-train the ResNet-18-PT model on the remaining subset of the ImageNet dataset. We have updated Appendix B with relevant details.
>
> **[References]**
>
> [1] Devlin, Jacob, et al. "BERT: Pre-training of Deep Bidirectional Transformers for Language Understanding." Proceedings of the 2019 Conference of the North American Chapter of the Association for Computational Linguistics: Human Language Technologies, Volume 1 (Long and Short Papers). 2019.
>
> [2] Vu, Tu, et al. "Exploring and Predicting Transferability across NLP Tasks." Proceedings of the 2020 Conference on Empirical Methods in Natural Language Processing (EMNLP). 2020.
>
> [3] Ramasesh, Vinay Venkatesh, et al. "Anatomy of Catastrophic Forgetting: Hidden Representations and Task Semantics." In International Conference on Learning Representations. 2021.
>
> [4] Abdelsalam, Mohamed, et al. "IIRC: Incremental Implicitly-Refined Classification." Proceedings of the IEEE/CVF Conference on Computer Vision and Pattern Recognition. 2021.

---

> ### Author Response · Authors · 2021-11-20
> **Response to Reviewer UX5R (Part 2/3)**
>
> **[Task similarity/dissimilarity]**
>
> For a fair comparison between pre-trained and randomly initialized models, we wanted to explicitly control for and remove the overlap between pre-training and downstream tasks. For NLP experiments, this is less of a concern as Transformer models are pre-trained on generic language modeling tasks. However, for CV, publicly available ResNet models are pre-trained on ImageNet which overlaps with CIFAR-100 in terms of class labels. Therefore, we make sure that the subset of the ImageNet corpus we use does not have any visually and semantically overlapping classes with the CIFAR-100 dataset (we have updated Appendix B with more details). Specifically, we pre-train a new ResNet-18 model on 733 subset classes of the original ImageNet corpus. In that case, all CIFAR-50/ CIFAR-100 tasks are likely going to be equally distant from the pre-training task.
>
> Furthermore, we agree that the analysis of the effect of downstream tasks similarity and dissimilarity on forgetting would be an interesting extension of our work, and we present some preliminary results on NLP datasets here. We believe, however, that this topic is out of scope for this paper, and we would not be able to do both this topic and our current investigation of forgetting justice in the limited space available.
>
> **Some preliminary results:** To study the effect of task similarity/dissimilarity on forgetting, we compute a task similarity measure between the given downstream tasks and study how it relates to the forgetting in the case of pre-trained BERT-base. For our analysis, we consider diverse tasks from 15-dataset-NLP. For each task, we compute a task embedding as a mean feature vector (BERT [CLS] representation from a pre-trained model) over all validation examples. This task embedding helps us capture domains (Wikipedia, Quora questions, etc.) of our tasks, and domain similarity is known to be a relevant factor for transfer [2]. Given a pair of task embeddings, we define the task similarity as cosine similarity between these embeddings. For example, when we compare the task similarity between BoolQ (Yes/No question answering task) and other tasks, we see it ranges from 0.60 (BoolQ and Decomp) to 0.97 (BoolQ and QNLI). We note that BoolQ/ QNLI examples are sourced from Wikipedia while Decomp examples are constructed from FactBank (new articles) and our task embeddings reflect these similarities/differences.
>
> Next, we consider 60 distinct pairs of tasks (Task1 → Task2) and compute the drop in performance (forgetting) of pre-trained BERT-base on Task1 after training on Task2. Surprisingly, we observe that the task similarity does not seem to be directly related to forgetting. Pairs of tasks with either relatively low (0.60-0.70) or high (0.95-0.99) similarity show less variation in forgetting [0.09-12.91] as compared to the forgetting between intermediate similarity tasks [1.88-28.42]. A similar observation is noted in the recent work on CV tasks [3] that the relation between task similarity and less forgetting is nuanced. We would like to note that for our analysis above, task similarity measure: (i) does not depend on the training labels (basically just consider domain information), (ii) it is symmetric in nature and we know that (positive/negative) transfer is asymmetric. Future works can explore other sophisticated similarity measures for further analysis [2].
>
> **[Paper presentation]**
>
> We believe that all relevant related works that are necessary for assessing our work are already discussed in the main manuscript (Sections 1, 4, 5, and 6). If the reviewer still feels that there are any that are missing, we are happy to address them over camera-ready. We have updated the captions for Tables 1, 2 with key findings and highlighted key results to enhance readability. We agree with the reviewer that we have summarized all our results into Table 1 and Table 2 and keep referencing them throughout the manuscript. We apologize for the confusion and welcome any suggestions that will help us further improve the paper presentation.
>
> **[Information on datasets and considered baselines]**
>
> In general, our decisions regarding the continual learning setup (task-incremental learning), baseline methods, benchmarks, and evaluation metrics, are informed by relevant recent works. This helps our analysis and evaluation of the discussed SAM method to be consistent with them. For detailed answers about the considered baselines and datasets, please refer to the **Response to Reviewer 6rM5 (Part 1/2) [Baselines]** and **Response to Reviewer 6rM5 (Part 2/2) [Datasets]**. We have updated the main text (Sections 2.2, 2.4) and Appendix B with more details.

---

> ### Author Response · Authors · 2021-11-20
> **Response to Reviewer UX5R (Part 1/3)**
>
> We thank the reviewer for their feedback and suggestions. We are glad that the reviewer acknowledges that: (i) empirical studies like ours that aim to identify and understand a phenomenon are necessary for the machine learning community, and (ii) our proposed solution based on empirical investigation leads to notable gains across several settings.
>
> *We have detailed responses to the reviewer's comments and look forward to hearing from the reviewer during the discussion period.*
>
> **[Novel Insights]**
>
> We agree with the reviewer that pre-trained models are known to provide robust gains when fine-tuned on a single downstream task [1]. However, in this paper, we study these models in the context of lifelong learning, specifically, the task incremental learning setting, where they are continuously fine-tuned on a sequence of tasks and undergo the forgetting phenomenon. We show that pre-trained models forget less compared to randomly initialized models, a phenomenon which to our knowledge has not been studied in any previous (non-contemporary) work. Furthermore, to explain this observation, we show that pre-trained models land in flat minima for all tasks in the sequences, compared to the randomly initialized models, which leads to less forgetting. Lastly, we show that explicitly optimizing for flat minima can further alleviate the forgetting for random and pre-trained initialization.
>
> The reviewer argues that some of the insights are well known or intuitive, so the work lacks novelty. However, we respectfully disagree with the reviewer that our main conclusions are not providing novel insights. We believe that the reviewer might not have considered a lifelong learning (sequential fine-tuning) setup that differentiates our analysis from vanilla single task fine-tuning setups. If the reviewer still believes that we are not novel, we respectfully ask that they provide pointers to non-contemporary published works with similar analysis/conclusions that we can use to position our claims appropriately.
>
> **[Comparison between pre-trained model and a randomly initialized model]**
>
> In the paper, we discuss our experimental design in detail (see Section 3). Ideally, we want sufficient training samples for each task so that we can ensure that both pre-trained and randomly initialized models under the same capacity can achieve comparable learning accuracy. Therefore, for NLP experiments in our study, we specifically selected those datasets where we have ample amount of data per task, e.g.: 115k training examples per task in 5-dataset-NLP and 279k training examples per task in Split YahooQA. From Table 1, we can clearly see that the learning accuracy (LA) for both these datasets across the initializations is comparable. On the other hand, for the commonly used CV benchmarks, we did not have enough training samples, for e.g., Split CIFAR-50, we just have 5000 training examples per task. Therefore, we ran our CV experiments with multiple epochs for better convergence of the randomly initialized models, thereby narrowing the gap between learning accuracy for both the initialization.
>
> **[Task ordering]**
>
> In the lifelong learning paradigm, one does not have upfront access to task orders. So, one of the desiderata of a lifelong learning method is to be robust to different task orderings. In fact, all of our experiments are run with 5 different task orderings (in Appendix B.1, we report task orderings/sequences used for our experimentation). We agree with the reviewer that several interesting experiments could be considered in this work, especially concerning the task order but we leave them for future research as our current focus here is to study/ develop methods that are agnostic to task orderings.

---

### Official Review · Reviewer_6rM5 · 2021-11-03

**Correctness:** 3
**Technical Novelty And Significance:** 3
**Empirical Novelty And Significance:** 3
**Recommendation:** 5
**Confidence:** 4

**Main Review:**

Pros
- The paper is very well-written and easy to follow.
The paper shows that pre-weights can alleviate catastrophic forgetting and suggests an explanation of it - pertained weights lead to wider minima. With that, the authors show that explicitly seeking flat basins during sequential fine-tuning results in even less forgetting. Again, the coherent structure and detailed analysis are a plus.

Concerns
-  Missing baselines.  Various methods have been proposed in the continual/lifelong learning line of work. Although it is hard to exhaustively evaluate all possible methods, some popular approaches are still missing, such as iCaRL and PackNet (parameter isolation-based approaches). Therefore, the authors may group their baselines based on method types, similar to the structures in the survey paper (e.g., https://arxiv.org/pdf/1909.08383.pdf).
- Limited setting. The paper focuses only on the Tabula rasa case, where incremental learners trained from scratch and do not require pretraining on large labeled datasets. A more realistic setting is to learn from pre-trained models. For example, a model is trained on the first 50 classes of CIFAR-100 and then incrementally learns the remaining classes. Is the paper trying to suggest CL benchmarks should use pre-trained weights instead of training from scratch?
- The CV datasets are relatively small.  The methods are evaluated on CIFAR datasets while in the literature (again checking the survey paper above),  larger datasets like ImageNet, iNaturalist, are used for evaluation.  Can the authors provide the analysis there?
- Other evaluation metrics, like Backward/Forward transfer?


**Summary Of The Paper:**

This paper provides an empirical study on pretraining for lifelong learning. The paper suggests that the generic pre-training implicitly alleviates the effects of catastrophic forgetting in lifelong learning and the reason is the pre-trained weights can ease forgetting by leading to wider minima. The model is evaluated on a range of datasets in CV and NLP to support the findings.





**Summary Of The Review:**

The paper provides an empirical study to understand the role of pre-training in lifelong learning.  Given that the paper draws conclusion based on empirical results, it's important to have a systematic study over the existing continual learning settings and baselines (e.g.,  various training/evaluation settings,  evaluation metrics such as Backward/Forward transfer,  datasets, etc.). The current manuscript pushes one step further towards understanding the role of pretraining, but still more studies can be added.

---

> ### Author Response · Authors · 2021-11-20
> **Response to Reviewer 6rM5 (Part 2/2)**
>
> **[Datasets]**
>
> We perform extensive experiments on the widely adopted standard benchmarks [1,2,3,4]. Particularly, for the CV domain, we use CIFAR-100 and 5-dataset. CIFAR-100 is more challenging and one of the realistic benchmarks because of the large number of tasks [2] (20 5-way classification tasks, 2.5k train examples/ task, 50k training examples in total). On the other hand, 5-dataset is a large dataset with 5 diverse 10-way classification tasks, CIFAR-10 (42.5k), MNIST (51k), Fashion-MNIST (9.57k), SVHN (62.27k), and notMNIST (15.52k), with a total of 180.9k training examples. The reviewer suggests conducting further analysis on larger datasets like ImageNet and iNaturalist (as used in [6]). Firstly, [6] uses Tiny ImageNet, a subset of 200 classes from the original ImageNet dataset, and constructs 10 20-way classification tasks, with 8k examples/ tasks, 80k training examples in total. We maintain that CIFAR-100 is comparable in size with Tiny ImageNet and has 20 tasks, making it a more challenging setup. Next, the iNaturalist dataset [6] spans 10 tasks with a varying number of classes and 156.2k training examples in total. Apart from the class imbalance nature of this dataset, it is similar in size to the 5-dataset. It is also not commonly used as a continual learning dataset. *We apologize for having missed the details about the 5-dataset size and have updated Appendix B with it*. Moreover, we also have results on diverse and large NLP datasets (Split-YahooQA has 1.4M training examples, 5-dataset-NLP has 575k training examples) and hardly any prior works evaluated on both CV and NLP benchmarks to showcase the generality of their findings.
>
> **[Evaluation metrics]**
>
> As our work primarily focuses on catastrophic forgetting, we compute the following metrics: **average accuracy**, **forgetting**, and **learning accuracy** to evaluate different methods [1,2,3,4]. Average accuracy measures the overall performance on all tasks after learning the last task in the sequence. Forgetting measures the average drop in the performance on all previously seen tasks after learning the last tasks. Note that **forgetting** is also referred to as **backward transfer** in the literature [7]. As models learn multiple tasks in the sequence, we hope that knowledge acquired during lifelong learning should aid the learning of new tasks (**forward transfer**). Learning accuracy measures the learning capability of the model to the new task and indirectly captures the notion of forward transfer.
>
> *Please let us know if our answers above address your concerns. We are happy to answer any follow-up questions!*
>
> **[References]**
>
> [1] Chaudhry, Arslan, et al. "On tiny episodic memories in continual learning." (2019).
>
> [2] Chaudhry, Arslan, et al. "Efficient Lifelong Learning with A-GEM." ICLR. 2019.
>
> [3] Mirzadeh, Seyed Iman, et al. "Understanding the Role of Training Regimes in Continual Learning." In Advances in Neural Information Processing Systems 33: Annual Conference on Neural Information Processing Systems 2020.
>
> [4] Prabhu, Ameya, et al. "Gdumb: A simple approach that questions our progress in continual learning." European conference on computer vision. Springer, Cham, 2020.
>
> [5] Hussain, Aman, et al. "Towards a robust experimental framework and benchmark for lifelong language learning." In Thirty-fifth Conference on Neural Information Processing Systems Datasets and Benchmarks Track (Round 1) 2021.
>
> [6] Delange, Matthias, et al. "A continual learning survey: Defying forgetting in classification tasks." IEEE Transactions on Pattern Analysis and Machine Intelligence (2021).
>
> [7] Lopez-Paz, David, and Marc'Aurelio Ranzato. "Gradient episodic memory for continual learning." Advances in neural information processing systems 30 (2017): 6467-6476.

---

> ### Author Response · Authors · 2021-11-20
> **Response to Reviewer 6rM5 (Part 1/2)**
>
> We thank the reviewer for their time and remarks on our work. We are pleased that the reviewer finds the paper as well-written, coherent, comprehensive, and easy to understand.
>
> In this work, our decisions regarding the continual learning setup (task-incremental learning), **baseline methods**, **benchmarks**, and **evaluation metrics**, are informed by relevant recent works [1,2,3,4]. This helps our analysis and evaluation of the discussed SAM method to be consistent with them.
>
> **[Baselines]**
>
> Based upon the suggestion of the reviewer, here’s a brief discussion about the baselines grouped by method types:
>
> 1. **Regularization-based methods:** alleviate forgetting by limiting updates to the parameters important to previously learned tasks. Elastic Weight Consolidation (**EWC**) is one of the representative approaches in this category and we consider it for our experimentation (in line with [1,2,3,4]).
>
> 2. **Replay-based methods:** augment the base model with an episodic memory module to retain limited examples from previous tasks. Different methods use these examples either for rehearsal (**Episodic Replay/ER** and iCaRL) or as constraints during the optimization process (GEM and **A-GEM**). The primary difference between iCaRL and ER is the nature of the task classifier: iCaRL uses the nearest neighbor (nonparametric) and ER uses Softmax (parametric) classifiers. In the class-incremental learning setup, we do not know the number of classes beforehand, therefore, iCaRL is more appealing and is a prominent baseline. On the other hand, for the task-incremental learning setup, we know the classes beforehand, therefore in line with the relevant works [1,2,3,4], we consider ER over iCaRL as a baseline. Also, the recent work [5] on lifelong language learning demonstrates that simple ER outperforms all of the previous methods under realistic settings, and therefore for all NLP experiments we compare with a state-of-the-art ER baseline.
>
> 3. **Parameter isolation-based methods:** assign different dedicated subsets of the model parameters to each task either by freezing them in a single network or dynamically adding new parameters per task. As discussed in [6], **PackNet** is one of the prominent methods in this category that iteratively prunes the fixed network to free up parameters after learning each task. Although such a method is shown to be robust to forgetting (almost zero forgetting, see Figure 2 in [6]), it does not scale to a large number of tasks [5] (runs out of the fixed capacity and is unable to learn new tasks in the sequence). Moreover, by keeping the weights frozen (freezing the previous task weights in PackNet), one is explicitly disabling the updates to the underlying model. We argue that such a method does not faithfully assess the fundamental challenge of learning continually, specifically catastrophic forgetting. On the contrary, fine-tuning-based methods update the model parameters and are prone to severe forgetting. We agree that continual learning is broader than catastrophic forgetting, however, in this work we decide to study the forgetting phenomenon in detail on one of the challenging setups, if not the most challenging.
>
> **Initialization/Optimization/Training dynamics:** Apart from the above-discussed taxonomy of the existing baselines, we discuss an emerging family of methods that view the catastrophic forgetting phenomenon through the lens of machine learning basics like network architecture, initialization, optimization or training dynamics, etc. Our work in this paper explains the inherent bias in pre-trained weights to implicitly alleviate forgetting. Further, **StableSGD** [3] shows that controlling the training dynamics: varying learning rate, drop, learning rate decay, and batch size, reduces forgetting. As discussed in the main text (Section 5), the procedure for tuning these hyper-parameters is ill-defined and expensive for lifelong learning of tasks, therefore rendering StableSGD less helpful. Moreover, it is unclear how to apply StableSGD to Transformers architectures and discrete NLP inputs. Nevertheless, we compare our discussed SAM optimization procedure with StableSGD on commonly used CV benchmarks (see Table 1).
>
> **[Setting]**
>
> In this work, we focus on evaluating generically available pre-trained models without any prior information about the downstream tasks. To explicitly control for this fact, we even made sure that our ImageNet pre-training corpus does not share any semantically similar labels with CIFAR-100 dataset (*we have updated Appendix B with more details*). Our findings suggest that pre-trained weights are better suited for continual learning and future works should consider developing methods inspired from them or building on top of them.
> *Let us know if we answer your question convincingly else we are happy to answer again if you could rephrase it a bit.*

---

### Author Response · Authors · 2021-11-25
**General response**

We thank our reviewers for their detailed comments and valuable suggestions. We are glad that the reviewers find: (i) our paper to be well-written, coherent, and easy to understand (**reviewer 6rM5**), (ii) our studies in this work to be useful for the broader community as large pre-trained models have become a norm (**reviewers UX5R, xCMa**), (iii) our approach to tackling catastrophic forgetting as being interesting and practical rather than fine-grained changes in existing approaches (**reviewer xCMa**), and (iv) our claims to be strongly supported with comprehensive experiments, including both qualitative and quantitative analyses (**reviewers 6rM5, UX5R, xCMa**).

We have provided a detailed response to each reviewer and uploaded the revised version of the paper. Specifically,
 1. We have updated section 2 with more discussions around benchmark datasets (section 2.2), evaluation metrics (section 2.3), and baseline methods (section 2.4).
2. In Appendix B (Datasets), we have listed steps to preprocess the ImageNet pre-training corpus to filter semantically similar classes with CIFAR-100.
3. In section 2.2 (datasets and task) and Appendix B.1 (Task Sequences), we report task orderings/sequences used for our experimentation.
4. We have added plots (Figure 3, 4) along with insightful analyses around task-specific results for 5-dataset and 5-dataset-NLP in Appendix C.
5. We have updated the captions for Tables 1, 2 with key findings and highlighted key results to enhance readability.
6. We have updated Table 3 with sharpness metrics for Finetune and Finetune + SAM methods (5-dataset and Split CIFAR-50).
7. We plot loss contours for Task 1/ Task 2 of Split CIFAR-50 (Figure 6) and 5-dataset (Figure 7), across four different methods: Finetune, Finetune + SAM, ER, and ER + SAM.

*We hope that our revisions and responses address all of your concerns. We are happy to answer any further questions.*

---

### Decision · Program_Chairs · 2022-01-20

**Decision:**

Reject

**Comment:**

The paper presents an empirical study on the impact of pertained model on lifelong learning. It concludes that the generic pertaining can benefit the lifelong learning duet the flatter loss landscape and evaluates on CV and NLP tasks. The paper is well written with detailed analysis. However, there is concerns on its limited setting and the conclusion is known in the community and based on empirical studies.